# *NAP*: *N*eural 3D *A*rticulated Object *P*rior

**Jiahui Lei[1]**     **Congyue Deng[2]**     **Bokui Shen[2]**     **Leonidas Guibas[2]**     **Kostas Daniilidis[13]**
[1] University of Pennsylvania     [2] Stanford University     [3] Archimedes, Athena RC
{leijh, kostas}@cis.upenn.edu, {congyue, willshen, guibas}@cs.stanford.edu

https://www.cis.upenn.edu/~leijh/projects/nap

## Abstract

We propose Neural 3D Articulated object Prior (NAP), the first 3D deep generative model to synthesize 3D articulated object models. Despite the extensive research on generating 3D static objects, compositions, or scenes, there are hardly any approaches on capturing the distribution of articulated objects, a common object category for human and robot interaction. To generate articulated objects, we first design a novel articulation tree/graph parameterization and then apply a diffusion-denoising probabilistic model over this representation where articulated objects can be generated via denoising from random complete graphs. In order to capture both the geometry and the motion structure whose distribution will affect each other, we design a graph denoising network for learning the reverse diffusion process. We propose a novel distance that adapts widely used 3D generation metrics to our novel task to evaluate generation quality. Experiments demonstrate our high performance in articulated object generation as well as its applications on conditioned generation, including Part2Motion, PartNet-Imagination, Motion2Part, and GAPart2Object.

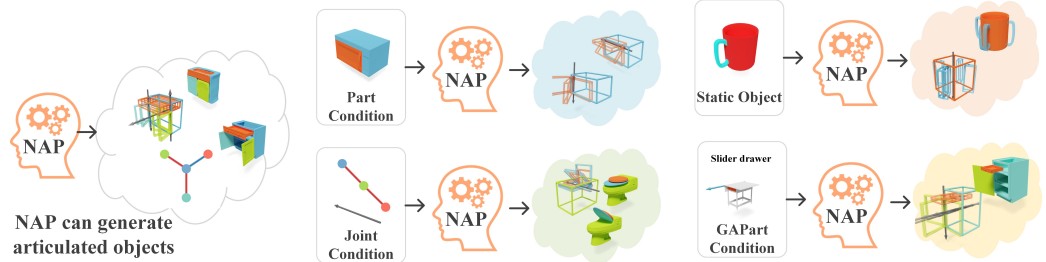

Figure 1: NAP can unconditionally generate articulated objects (left). It can be conditioned on just parts or joints (mid), a subset of parts plus joints, or over-segmented static objects (right).

## 1   Introduction

Articulated objects are prevalent in our daily life. As humans, we have strong prior knowledge of both object part geometry and their kinematic structures. Such knowledge is most heavily leveraged when a designer designs a cabinet from scratch, creating both its geometry and motion structure. For learning systems, an interesting challenge is to capture such priors as a generative model that can synthesize articulated objects from scratch. While there has been extensive research on generative models for static 3D objects [1–10], compositions [11–18], and scenes [19–30], the study of priors regarding closely linked 3D part geometry and 3D motion structures has been relatively neglected. In this work, we study *how to synthesize articulated objects*, i.e., how to generate a full description

37th Conference on Neural Information Processing Systems (NeurIPS 2023).

of an articulated object, an actual URDF [31], including how each part looks like, which pairs of parts are connected, and with what kind of joint. In contrast to static 3D generation, generating articulated objects involves modeling not only the distribution and composition of geometry but also the motion structures that determine the possible relative movements between rigid parts. Our task is also different than 3D human/hand synthesis [32–35] where the articulation structure is given and only the degrees of freedom are generated when predicting human poses/sequences. The generation of articulation can further impact simulation and mechanism design and be useful for inference when conditioned on either geometry or kinematic structure.

However, there are several challenges for articulated object generation. Existing datasets of articulated objects contain highly irregular data since articulated objects have different numbers of parts, diverse part connectivity topology, and different joint motion types. In order to enable efficient processing of diverse geometry and structures by a neural architecture, we propose a novel unifying articulation tree/graph parameterization (Sec. 3.1) to represent articulated objects. We take advantage of recent progress in diffusion models for 3D generation and develop a diffusion-denoising probabilistic model over our parameterization to faithfully model the irregular distribution of articulated objects (Sec. 3.2). Since we are modeling the joint distribution of the part geometry and the inter-part motion constraints, we design a graph denoising network (Sec. 3.3) to gradually exchange and fuse information between the edge and nodes on our articulation graph.

To evaluate articulated object generation quantitatively, we adopt the widely used 3D shape generation metric for articulated objects by introducing a novel distance measure between two articulated objects. Through extensive comparisons, we demonstrate high performance in articulated object synthesis and further conduct several ablations to understand the effect of each system component. Using the learned prior knowledge, we demonstrate conditioned generation applications, including Part2Motion, PartNet-Imagination, Motion2Part, and GAPart2Object .

In summary, our main contributions are **(1.)** introducing the articulated object synthesis problem; **(2.)** proposing a simple and effective articulation tree parameterization sufficiently efficient for a diffusion denoising probabilistic model that can generate articulated objects; **(3.)** introducing a novel distance for evaluating this new task; and **(4.)** demonstrating our high performance in articulated object generation and presenting several conditioned generation applications.

## 2    Related Work

**Articulated object modeling**. Modeling articulated objects has been a prolific field with existing works broadly classified into the categories of estimation, reconstruction, simulation, and finally, our generation. Note that there is a wider literature on semi-nonrigid objects, for example, human body [36], hands [37], and animals [38]. This paper focuses on everyday articulated objects, like the ones in PartNet-Mobility [39], which have more diverse and complex structures and are strictly multi-body systems. Estimation focuses on predicting the articulation *joint states (joint angles and displacements)* or *joint parameters (type, axis and limits)* from sensory observations, using approaches ranging from probabilistic models [40–44], interactive perception [45–51] and learning-based inference [52–65]. Reconstruction of articulated objects focuses on reconstructing both articulation and geometry properties of the objects, using techniques ranging from structure from motion [66], learning-based methods [67–69] and implicit neural representations [70–74]. While these methods mainly focus on surface reconstruction and joint states/parameter accuracy or are limited to pre-defined simple kinematic structures, we explicitly model the diverse and complex articulation motion structures. One closely related work to ours is [62], which predicts the joint parameters of articulated objects in PartNet [75] by training on PartNet-Mobility [39]. However, [62] is a single-point regression that does not capture the generative distribution of joints or parts. A growing field in robotics and embodied AI is building interactive environments that support physical interactions between the robot and the scene consisting of articulated objects [75, 39, 76–82]. Differing from all the above approaches, we build a **generative** prior of articulated objects, which extends beyond estimation and reconstruction. Such learned prior hasential further to accelerate the creation of realistic interactive 3D assets.

**Generative models for structures**. Generating structured articulated objects is closely related to generative models for structured data [83]. Part-based 3D object generation [11–18] has been widely studied with the main modeling target being the hierarchy of static rigid parts with or without a semantic label. Scene layout generation [19–30] utilizes compact scene parameterizations for indoor scene synthesis, where diffusion models on scene graphs have been recently introduced in [21, 22]. The generative diffusion approach has also been applied widely to 2D floor plans [84], protein struc-

tures [85], and graphs [86] .etc. Unlike all the existing works, our work focuses on modeling a new type of target – articulation structure plus part shapes, which requires joint reasoning between 3D geometry and kinematic structure.

**Diffusion models in 3D**. We mainly focus on the literature on diffusion models for 3D shape and motion generation. **Shape:** Diffusion models show impressive results in generating point clouds [1–3], meshes [87], implicit surfaces [4–10], neural radiance fields [88–91], or 4D non-rigid shapes [92]. However, these methods mostly focus on the single object level shape quality and do not pay attention to the kinematic structure. **Motion:** Diffusion models have seen many recent applications in motion generation given an articulation model. Such works generate text-conditioned human motion [32–35], and physically-viable [93], audio-driven [94, 95], scene-aware [96], multi-human [97] or animation [98] trajectories. Diffusion models for motion have also been applied to trajectory planning [99], visuomotor control [100], and rearrangement tasks [101]. Different from ours, existing works in motion diffusion rely on known geometries with known motion structures. We instead jointly model geometry and motion structure priors to create articulation models.

# 3 Method

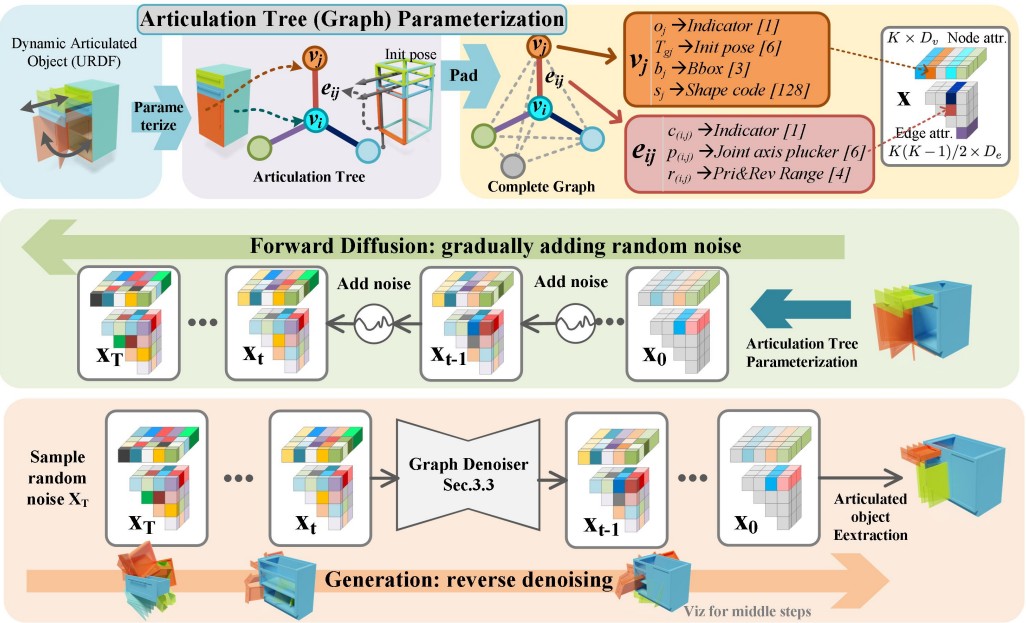

Figure 2: **Method**: **Parameterization (Top)**: We parameterize the articulated object as a tree, whose nodes are rigid parts and edges are joints; we then pad the tree to a complete graph of maximum node number and store it in the articulation graph attribute list $\mathbf{x}$. **Forward Diffusion (Middle)**: The parameterized attribute list $\mathbf{x}$ is gradually diffused to random noise. **Generation (Bottom)**: A Graph Denoiser (Fig 3, Sec. 3.3) samples a random articulation graph $\mathbf{x}_T$, gradually removes noise, and finally predicts $\mathbf{x}_0$. An object extraction stage (Sec. 3.2) including a minimum-spanning-tree algorithm is applied to the generated graph in the end to find the kinematic tree structure and the output articulated object.

We learn the articulated object priors using a diffusion model. However, an articulated structure must first be parameterized into a vector that can be the target of the diffusion. We introduce the parameterization of the articulated object as a tree (graph) in Sec. 3.1; the diffusion model on such parameterization in Sec. 3.2; and the denoising network in Sec. 3.3. An overview of our method is shown in Fig. 2.

## 3.1 Articulation Tree Parameterization

**Graph-based representation**. We follow the natural articulated object parameterization as in URDF [31] format where each object is defined by a graph with nodes being the parts and edges being the articulation joints. We make two assumptions: (1) *Tree assumption:* We assume no kinematic motion loop (cycle) exists in the graph and that the graph is connected. (2) *Screw joints:* We assume

each edge is a screw [60] with at most one prismatic translation and one revolute rotation. Most real-world articulated objects [39, 76, 82, 102] satisfy the above assumptions and their limitations are further discussed in Sec. 5.

**Nodes**. As shown in Fig. 2-top, we represent each rigid part as one node in the tree. A node captures its rigid part shape by (1) a shape latent code $s_i \in \mathbb{R}^F$ (with $F$ being channels, $i$ being the part index) that can decode the SDF [103] of the part surface, and (2) the bounding box edge lengths $b_i \in \mathbb{R}^3$ that can scale the decoded SDF properly into the part's scale. We pre-train an SDF shape Auto-Encoder [104, 103] for obtaining $s_i$, further details are provided in our supplementary material. Importantly, one should also specify how to assemble these parts into an object before further modeling the articulation motion. We do so by adding to the node attributes a part initial rigid transformation $T_i \in SE(3)$ in the global object frame representing the part initial configuration. $T_i$ comprises an axis-angle rotation and a translation, which can be written as a 6-dimensional vector. As articulated objects can be instantiated to different configurations so any of these configurations can be used as $T_i$, in practice, we consistently choose $T_i$ by exploiting the canonicalization in the large annotated dataset: we use the part poses in their rest states (zero joint angles and displacements) for $T_i$. Since we observe current datasets often align their rest states consistently to static un-articulated objects, for example, PartNet-Mobility [39] aligns with static PartNet [75]. To model the varying number of parts across objects, we define a maximum number of $K$ parts as well as a per-part binary indicator $o_i \in \{0, 1\}$ of part $i$ existence. In summary, a node $i$ has an attribute vector $v_i = [o_i, T_i, b_i, s_i]$ with dimension $D_v = 1 + 6 + 3 + F$, and the overall node component of an object graph is a feature of dimension $K \times D_v$.

**Edges**. The edge in the graph represents the motion constraint of each articulation joint. A joint possesses a 3D axis (a directed line), around which the revolute joint can rotate and along which the prismatic joint can translate. Inspired by [60], such a joint axis is represented by Plücker coordinates $(l \in \mathbb{S}^2, m \in \mathbb{R}^3)$. Here the 3D directional vector $l$ is from a unit sphere $\mathbb{S}^2$ and the momentum $m$ is perpendicular to $l$. Such representation avoids defining additional local joint coordinate frames with ambiguity (e.g. translating the joint coordinate frame along a revolute axis leads to equivalent joints). Such 6-dimensional Plücker coordinates $p_{(i,j)} = (l_{(i,j)}, m_{(i,j)})$ for the joint from part $i$ to $j$ are defined in the global object frame when parts are in their initial rest configuration $T_i, T_j$. To fully define the joint motion constraint, we also incorporate two joint state ranges $r_{(i,j)} \in \mathbb{R}^{2 \times 2}$ for both the prismatic translation and the revolute angle components (the left limit of the range can be negative). A purely prismatic joint will have its revolute component range set to $[0, 0]$ and vice versa. Following the node padding, we also pad the edges to a complete graph and use an indicator $c_{(i,j)}$ denoting the edge existence. Note that the above joint axis and range have a parent-child direction from part $i$ to $j$. When the direction flips, a notable benefit of expressing $p_{(i,j)}$ in the global object frame, rather than the local part frame, is the inherent relationship $p_{(i,j)} = -p_{(j,i)}$ and $r_{(i,j)} = r_{(j,i)}$, which motivates us to model the padded graph with only $K(K - 1)/2$ edges for all $i < j$ pairs. Since the nodes have no specific order, to avoid the sign flipping in $p_{(i,j)} = -p_{(j,i)}$ when permuting the nodes and to help the network learn a more stable prior, we explicitly embed the sign inside $p_{(i,j)} = -p_{(j,i)}$ into the edge existence indicator $c_{(i,j)} \in \{-1, 0, +1\}$. Here 0 indicates the non-existence of an edge and $+1, -1$ indicate the chiralities of existing edges, leading to $\hat{p}_{(i,j)} = \hat{p}_{(j,i)}$ (where $\hat{p}$ denotes the actual prediction target). In essence, each edge is characterized by an attribute vector $e_{(i,j)} = [c_{(i,j)}, p_{(i,j)}, r_{(i,j)}]$ with a dimension of $D_e = 1 + 6 + 4$. The overall edge parameterization of an object has shape $K(K-1)/2 \times D_e$. We will see later in Sec. 3.2 how to extract an articulated object model from the diffusion model prediction via the Minimum Spanning Tree.

### 3.2 Diffusion-Based Articulation Tree Generation

Our goal is to learn the distribution of articulated objects parameterized by the complete articulation graphs $\mathbf{x} = (\{v_i\}, \{e_{(i,j)}\}) \in \mathbb{R}^{KD_v + K(K-1)D_e/2}$. We apply a diffusion denoising probabilistic model [105] directly over the distribution of $\mathbf{x}$.

**Forward diffusion**. Given an articulation graph $\mathbf{x}_0$ from an object distribution $q(\mathbf{x}_0)$, we gradually add Gaussian noise with variance schedule $\beta_1 < \cdots < \beta_T$ and obtain a sequence $\mathbf{x}_1, \cdots, \mathbf{x}_T$ ended with a Gaussian distribution $p(\mathbf{x}_T) = \mathcal{N}(\mathbf{x}_T; \mathbf{0}, \mathbf{I})$. The joint distribution of the forward process is:

$$q(\mathbf{x}_{1:T}|\mathbf{x}_0) := \prod_{t=1}^{T} q(\mathbf{x}_t|\mathbf{x}_{t-1}), \quad q(\mathbf{x}_t|\mathbf{x}_{t-1}) := \mathcal{N}(\sqrt{1 - \beta_t}\mathbf{x}_{t-1}, \beta_t\mathbf{I}). \tag{1}$$

A notable property is that $\mathbf{x}_t$ at arbitrary timestep $t$ can be directly sampled from $\mathbf{x}_0$ with

$$q(\mathbf{x}_t|\mathbf{x}_0) = \mathcal{N}(\sqrt{\bar{\alpha}_t}\mathbf{x}_0, (1-\bar{\alpha}_t)\mathbf{I}) \tag{2}$$

where $\alpha_t := 1 - \beta_t$ and $\bar{\alpha}_t := \prod_{s=1}^{t} \alpha_s$.

**Reverse process**. Starting from a standard Gaussian distribution $\mathbf{x}_T \sim \mathcal{N}(\mathbf{0}, \mathbf{I})$, we aim to learn a series of Gaussian transitions $p_\theta(\mathbf{x}_{t-1}|\mathbf{x}_t)$ parameterized by a neural network with learnable weight $\theta$ that gradually removes the noise. The distribution of the reverse process is:

$$p_\theta(\mathbf{x}_{0:T}) := p(\mathbf{x}_T)\prod_{t=1}^{T} p_\theta(\mathbf{x}_{t-1}|\mathbf{x}_t), \quad p_\theta(\mathbf{x}_{t-1}|\mathbf{x}_t) := \mathcal{N}(\mu_\theta(\mathbf{x}_t, t), \Sigma_\theta(\mathbf{x}_t, t)). \tag{3}$$

Following [105], we set $\Sigma_\theta(\mathbf{x}_t, t) = \sigma_t^2\mathbf{I}$ and model this reverse process with Langevin dynamics

$$\mathbf{x}_{t-1} = \frac{1}{\sqrt{\alpha_t}}\left(\mathbf{x}_t - \frac{1-\alpha_t}{\sqrt{1-\bar{\alpha}_t}}\epsilon_\theta(\mathbf{x}_t, t)\right) + \sigma_t\mathbf{z}, \quad \mathbf{z} \sim \mathcal{N}(\mathbf{0}, \mathbf{I}) \tag{4}$$

where $\epsilon(\mathbf{x}_t, t)$ is a learnable network approximating the per-step noise on $\mathbf{x}_t$.

**Training objective**. We optimize the variational bound on the negative log-likelihood

$$L := \mathbb{E}_q\left[-\log\frac{p_\theta(\mathbf{x}_{0:T})}{q(\mathbf{x}_{1:T}|\mathbf{x}_0)}\right] = \mathbb{E}_q\left[-\log p(\mathbf{x}_T) - \sum_{t\geq 1}\log\frac{p_\theta(\mathbf{x}_{t-1}|\mathbf{x}_t)}{q(\mathbf{x}_t|\mathbf{x}_{t-1})}\right] \geq \mathbb{E}[-\log p_\theta(\mathbf{x}_0)] \tag{5}$$

With Eq. 4, the objective simplifies to

$$\mathbb{E}_{\mathbf{x}_0, \epsilon}\left[\frac{\beta_t^2}{2\sigma_t^2\alpha_t(1-\bar{\alpha}_t)}\left\|\epsilon - \epsilon_\theta(\sqrt{\bar{\alpha}_t}\mathbf{x}_0 + \sqrt{1-\bar{\alpha}_t}\epsilon, t)\right\|^2\right] \tag{6}$$

*w.r.t.* learnable network $\epsilon_\theta$. We refer the readers to [105] for more details.

**Output Extraction**. The above-described Euclidean space denoising intermediate steps as well as the final output may not strictly lie on the manifold of articulated objects parameterized as in Sec. 3.1 (e.g. the generated joint Plücker coordinates may not be valid because $l$ and $m$ may not be orthogonal). We found that utilizing the diffusion as in Euclidean space and projecting the generated $\mathbf{x}$ back to a valid articulation graph in the final step already leads to practically good results. As mentioned in Sec. 3.1, once we have completed the denoising process, a post-processing step is applied to obtain the final articulated object model from the generated $\mathbf{x}$. First, we identify the existing nodes by the generated nodes indicator $o$ with a threshold and make sure there are at least two foreground nodes. Then we use the predicted edge chirality $|c|$ as the edge value to find the minimum spanning tree in the generated graph as the output tree topology. As in Sec. 3.1, we model half of the edges in the complete graph with node index $i < j$ and put the edge direction explicitly to the chirality sign. During the object extraction stage, if $c < 0$, we flip the predicted joint direction on this edge and if $c > 0$ we keep the joint direction, which is equivalent to flipping the parent and child order of this edge when $c < 0$. Finally, the predicted joint coordinates are projected from $\mathbb{R}^6$ to Plücker coordinates by normalizing the predicted $l$ to unit vector and subtracting the parallel component of $m$ on $l$ to make predicted $m$ orthogonal to $l$. We decode the part shape code to an SDF and extract the part mesh via marching cubes. We alternatively can retrieve the nearest part in the training set as most scene generation methods do [23, 22].

**Conditioned Generation**. A favorable property of diffusion models is that conditions can be directly added to the inference processes using Bayes' rule without any modification to the training process. Here we perform conditioned generation by fixing the known part of variable $\mathbf{x}$ as in the image inpainting [106] and shape completion works. For a variable $\mathbf{x} = m \odot \mathbf{x}^{\text{known}} + (1-m) \odot \mathbf{x}^{\text{unknown}}$ ($\odot$ means element-wise multiplication) with known entries $\mathbf{x}^{\text{known}}$ fixed unknown entries $\mathbf{x}^{\text{unknown}}$ to be completed which are separated by mask $m$, we can sample the known entries directly following Eq. 2 by adding Gaussian noise to the known input and generating the unknown entries using reverse diffusion,

$$q(\mathbf{x}_{t-1}^{\text{known}}|\mathbf{x}_0) = \mathcal{N}(\sqrt{\bar{\alpha}}\mathbf{x}_0, (1-\bar{\alpha}_t)\mathbf{I}), \quad p_\theta(\mathbf{x}_{t-1}^{\text{unknown}}|\mathbf{x}_t) = \mathcal{N}(\mu_\theta(\mathbf{x}_t, t), \Sigma_\theta(\mathbf{x}_t, t)). \tag{7}$$

Since our method directly diffuses in graph space, we can apply precise control of the parts and joints condition, enabling, thus, a disentanglement. In Sec. 4.4, we will show applications with different $\mathbf{x}^{\text{known}}$ and $\mathbf{x}^{\text{unknown}}$.

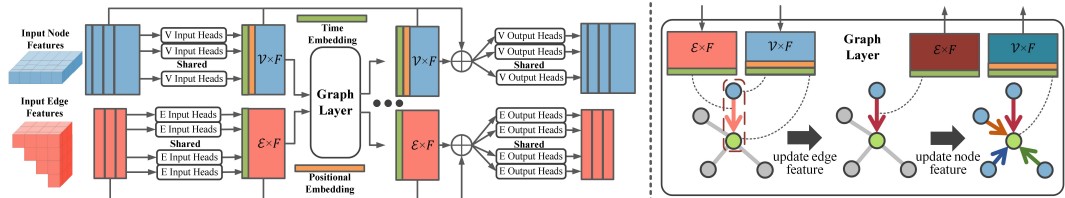

Figure 3: Network architecture. Left: input the node and edge list of a noisy articulation graph, a stack of graph layers will fuse and exchange information on the graph and output the noise that has to be removed. Right: details in the graph layer.

### 3.3 Denoising Network

**Network architecture**. Since $\mathbf{x}$ represents both the part geometry and the joint motion constraints, we utilize a graph denoising network as shown in Fig. 3, which exchanges and fuses information between parts and joints. The network inputs a noisy articulate graph $\mathbf{x}$ and outputs noise in the same shape as $\mathbf{x}$. The input attributes for every node are first encoded by input heads shared across nodes to a list of node features $\{f_i\}$. Similarly, edge attributes are encoded to a list of edge features $\{g_{(i,j)}\}$. Then the node and edge feature lists are updated via the graph layers (Fig. 3 right). Finally, all hidden features, including the input attributes, are concatenated together and decoded to the outputs via shared output heads. Note that the network is shared across all time steps through denoising, and the time step is encoded and concatenated to the hidden features. Similar to recent work on scene graph generation [22], we also append the positional encoding of the part instance id to the node features to provide stronger guidance in the early denoising stages when the part pose information is ambiguous. To learn with positional encoding, we randomly permute the nodes' order during training.

**Graph layer**. The key building block of the denoising network is the graph layer shown in Fig. 3 right. The edge feature is first updated via an edge MLP by fusing the input edge feature $g_{(i,j)}$ and neighboring nodes $i, j$'s features: $g'_{(i,j)} = \text{MLP}(f_i, f_j, g_{(i,j)})$. Then, we aggregate the updated edge features to the nodes by attention weights. We compute the query $Q(f_i)$ and key $K(f_i)$ from the input node features via two MLPs and use their inner product as the attention weights; the graph attention update of the node $i$ is: $f'_i = \sum_{j=1}^{K} \text{softmax}_j(Q(f_i)^T K(f_j)) g'_{(i,j)}$. We additionally do a PointNet [107]-like global pooling over all the graph node features after the attention aggregation to capture more global information.

## 4 Experiments

We examine 4 important questions with our experiments: (1) How can we evaluate articulated object generation? (Sec. 4.1) (2) How well does NAP capture the distribution of articulated objects? (Sec. 4.2) (3) How effective is each of NAP's components? (Sec. 4.3) (4) What applications can NAP enable? (Sec. 4.4)

### 4.1 Evaluation Metrics

Since we are the first to study articulated objects in a generative setting, we propose a new distance metric between two articulated objects for adopting widely used shape generation metrics. Since a generated object's shape and motion structure are dependent, we can not evaluate them separately. Such a challenge is poorly addressed in existing works. In articulated object modeling, existing works either consider a fixed kinematic structure [71, 70] or a given geometry [108]. In graph generation works [109–111], structures are examined by themselves without need to measure geometry. Thus, we propose a new distance metric, **Instantiation Distance (ID)**, to measure the distance between two articulated objects considering both the part geometry and the overall motion structure.

We treat an articulated object $O$ as a template that, given the joint states $q \in \mathcal{Q}_O$ in object's joint range $\mathcal{Q}_O$, it returns the overall articulate mesh $\mathcal{M}(q)$ and the list of part poses $\mathcal{T}(q) = \{T_{\text{part}} \in SE(3)\}$. We compute the distance between two articulated objects in different joint states by

$$\tilde{d}(O_1, q_1, O_2, q_2) = \min_{T_i \in \mathcal{T}_1(q_1), T_j \in \mathcal{T}_2(q_2)} \left\{ D(T_i^{-1} \mathcal{M}_1(q_1), T_j^{-1} \mathcal{M}_2(q_2)) \right\}, \tag{8}$$

Table 1: Articualted object synthesis comparison with Instantiation Distance

| Generative Paradigm/Method | Part SDF Shape | | | Part Retrieval Shape | | |
|---|---|---|---|---|---|---|
| | MMD ↓ | COV ↑ | 1-NNA ↓ | MMD ↓ | COV ↑ | 1-NNA ↓ |
| Auto-Decoding (StructNet) | 0.0435 | 0.1871 | 0.8820 | 0.0390 | 0.2316 | 0.8675 |
| Variational Auto-Encoding (StructNet) | 0.0311 | 0.3497 | 0.8085 | 0.0289 | 0.3363 | 0.7918 |
| Autoregressive (ATISS-Tree) | 0.0397 | 0.3808 | 0.6860 | 0.0333 | 0.4120 | 0.6782 |
| Latent Diffusion (StructNet) | 0.0314 | 0.4365 | 0.6269 | 0.0288 | 0.4477 | 0.6102 |
| Articulation Graph Diffusion (Ours) | **0.0268** | **0.4944** | **0.5690** | **0.0215** | **0.5234** | **0.5412** |

where $T_i^{-1}\mathcal{M}_1(q_1)$ means canonicalizing the mesh using its $i$th part pose, and $D$ is a standard distance that measures the distance between two static meshes. Specifically, we sample $N = 2048$ points from two meshes and compute their Chamfer Distance. Intuitively, the above distance measures the minimum distance between two posed articulated objects by trying all possible canonicalization combinations. Then, we define the instantiation distance between $O_1$ and $O_2$ as:

$$ID(O_1, O_2) = \mathbb{E}_{q_1 \in \mathcal{U}(\mathcal{Q}_{O_1})} \left[ \inf_{q_2 \in \mathcal{Q}_{O_2}} \left( \tilde{d}(O_1, q_1, O_2, q_2) \right) \right]$$
$$+ \mathbb{E}_{q_2 \in \mathcal{U}(\mathcal{Q}_{O_2})} \left[ \inf_{q_1 \in \mathcal{Q}_{O_1}} \left( \tilde{d}(O_1, q_1, O_2, q_2) \right) \right], \tag{9}$$

where $q \in \mathcal{U}(\mathcal{Q}_O)$ means uniformly sample joint poses from the joint states range. The instantiation distance measures the two-side expectation of minimum distance to the other object over all possible joint configurations. However, the $\inf$ inside the expectation requires expensive registration between two articulated objects so it is non-trackable in practice when computing all distance pairs between the reference and sampled object sets. In practice, we approximate the above distance by uniformly sampling $M$ joint poses $Q_1 = \{q_k | q_k \in \mathcal{U}(\mathcal{Q}_{O_1}), k = 1, \dots, M\}$ and the approximated distance is:

$$ID(O_1, O_2) \approx \frac{1}{M} \sum_{q_1 \in Q_1} \left[ \min_{q_2 \in Q_2} \left( \tilde{d}(O_1, q_1, O_2, q_2) \right) \right] + \frac{1}{M} \sum_{q_2 \in Q_2} \left[ \min_{q_1 \in Q_1} \left( \tilde{d}(O_1, q_1, O_2, q_2) \right) \right], \tag{10}$$

and we set $M = 10$ in our ID for all evaluations. This pairwise distance can be plugged into the standard metrics for shape generation. Specifically, we adopt the following three metrics [112] for our evaluation: **minimum matching distance (MMD)** that measures the generation quality, **coverage (COV)** that tests the fraction the reference set is covered, and **1-nearest neighbor accuracy (1-NNA)** that measures the distance between the two distributions by 1-nn classification accuracy.

## 4.2 Articulated Object Synthesis

**Baselines**. We adapt existing models in related tasks and compare our method with them. Specifically, we adapt architecture designs from semantic-part-based shape generation [11] and scene-graph-based indoor scene synthesis [23], and equip them with different generative paradigms including auto-decoding [113], VAE [11], autoregressive models [23], and latent diffusion [4, 5]. We refer to our supplementary for adaptation and details of these baselines.

**Setups**. We train all the methods on PartNet-Mobility [39] across all categories jointly, with a maximum 8 rigid parts ($K = 8$) and a train-val-test split ratio $[0.7, 0.1, 0.2]$. Since the dataset does not include parts orientation and all initial part poses have rotation $\boldsymbol{I}$, we ignore the rotation in the parameterization (Sec. 3.1) for all the methods. For a fair comparison, all the methods are controlled to a similar number of learnable parameters. We evaluate the generated articulated object models with both the reconstructed meshes from the generated shape code and the retrieved part meshes from the training set.

**Comparison**. We report quantitative results in Tab. 1, and qualitative comparisons in Fig. 4. While the Auto-Decoding baseline fits the training set, its generation performance is poor when sampling in the latent space as shown in Fig. 4-AD, which suggests a weak regularization in the latent space. Using VAE to replace auto-decoding brings about better regularization of the latent space, resulting in an increase in all metrics. Sampling from the prior of VAE also leads to more meaningful generations, as shown in Fig. 4-VAE. For the autoregressive ATISS-Tree baseline, we see a relative

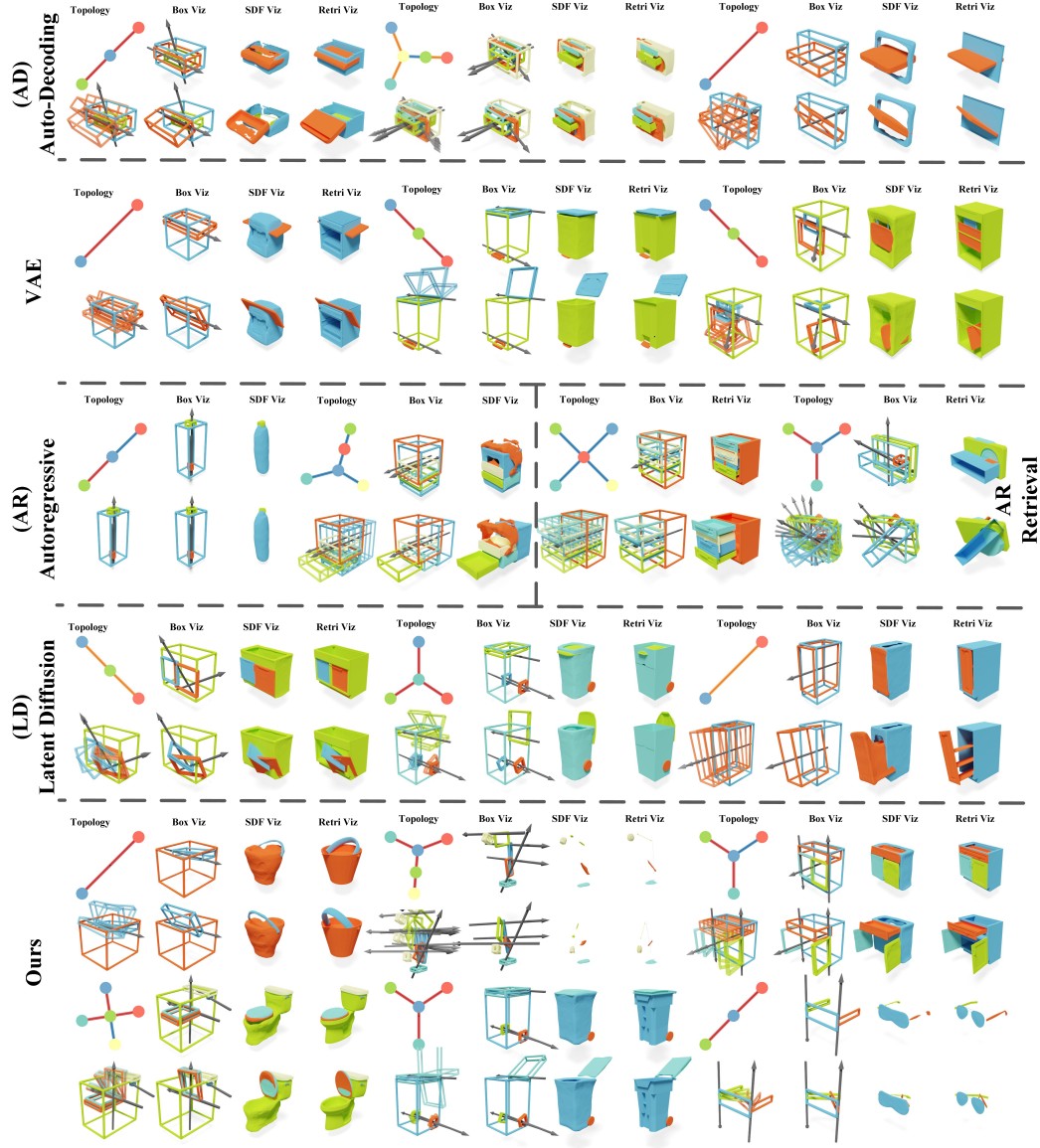

Figure 4: **Articulated object generation results.** Each generated object is visualized with (1) graph topology (top left), where the edge color means blue–prismatic, red–revolute, and orange–hybrid; (2) the predicted part bounding boxes and joints under different joint states (second column), and the overlay of multiple states reflecting the possible motion (bottom left); (3) reconstructed part meshes from the generated shape code (third column); (4) retrieved part meshes (right column).

increase in a majority of metrics compared to VAE. Part shape retrieval leads to further improvement in both motion structure and part shape since using retrieval at each autoregressive step can decrease the deviation from the training distribution. Interestingly, as shown in Fig. 4, we observe that the autoregressive method also has a tendency to append too many nodes to the tree, resulting in overlapping parts. Latent diffusion works the best among our baselines. We hypothesize that it is due to the superiority of the diffusion model as a sampler in the latent space, mapping the prior Gaussian to reliable regions where the trained decoder performs well near training samples. However, as the generation happens in the latent space and the generated latent code have to be decoded, slight error or changes in the latent space may lead to unrealistic or wrong articulations, which is shown in Fig. 4-LD. Different from the latent diffusion, our method directly applies diffusion in the articulation tree space, which can generate diverse and high-quality articulation models and achieves a better performance comparing to baselines.

Table 2: Ablation studies with Instantiation Distance

| Ablation | Part SDF Shape | | | Part Retrieval Shape | | |
|---|---|---|---|---|---|---|
| | MMD ↓ | COV ↑ | 1-NNA ↓ | MMD ↓ | COV ↑ | 1-NNA ↓ |
| Full | **0.0268** | **0.4944** | 0.5690 | **0.0215** | 0.5234 | **0.5412** |
| No PE | 0.0282 | 0.4766 | **0.5490** | 0.0227 | **0.5457** | 0.5557 |
| No Attn. on Edge | 0.0286 | 0.4766 | 0.5668 | 0.0232 | 0.5234 | 0.5568 |
| No Graph Conv | 0.0331 | 0.4432 | 0.6570 | 0.0281 | 0.4722 | 0.6481 |

### 4.3 Ablation Studies

We verify our denoising network design by ablating components in our full network, and the comparison is shown in Tab. 2. Specifically, we examine the effectiveness of positional encoding, attention weights on edges, and graph convolutions. First, removing the positional encoding will slightly worsen performance, since its presence can help to distinguish the nodes at the early stages of reverse diffusion [22]. Second, replacing attention weights with mean pooling when aggregating the neighboring nodes' information results in a drop in performance. Finally, we justify the importance of node-edge information exchange in the graph convolution by removing the graph convolution layer and replacing it with a PointNet [107]-like global pooling layer, where the connectivity of the graph is ignored and the information exchanges through a symmetric global pooling. We observe a larger performance decrease, which justifies our graph-based processing.

### 4.4 Applications with Conditioned Generation

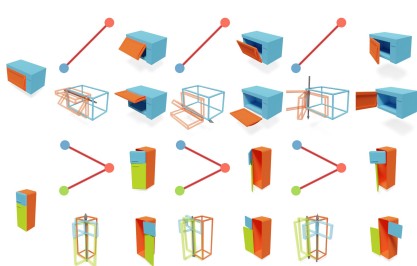

Figure 5: **Part2Motion**: Known part condition on the left, diverse motion proposals on the right.

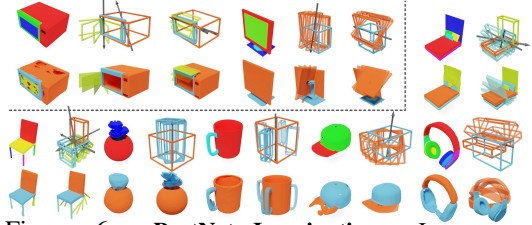

Figure 6: **PartNet Imagination**: Input over-segmented static PartNet shapes (top left) are grouped into rigid parts (bottom left) and be hallucinated with articulations (right). Both training categories (encircled by dashed lines) and out-of-distribution objects are shown.

Once trained, our method can be used directly for conditioned generation. Following Sec. 3.2, we demo applications conditioning on various known attributes in **x**.

**Part2Motion**. We first show that when knowing the static part attributes, how NAP can suggest motion structures, i.e., with $\mathbf{x}^{\text{known}} = \{v_i\}$ and $\mathbf{x}^{\text{unknown}} = \{e_{(i,j)}\}$ in Eq. 7. We use the unseen object part attributes from the test set as conditions and the generated motion structure is in Fig. 5. We observe diverse and plausible motion suggestions that cover the ambiguity of the closed doors.

**PartNet Imagination**. NAP uses PartNet-Mobility [39] for training, which is only a small subset of the large-scale but static PartNet [75]. We show that NAP can be used to imagine possible motion structures in static PartNet [75]. Starting from the finest semantic part labeled in PartNet [75] (as an over-segmentation), we use a simple contrastive learned grouping encoder (see Suppl.) to group the fine semantic parts into rigid parts and then apply the same node condition as in Part2Motion. Fig. 6 shows our predictions. Interestingly, we find the learned articulation prior can be applied to out-of-distribution object categories and generate intriguing motion structures that adhere to some human recognizable logic (e.g., the mug in Fig. 6 bottom, where the handle rotates around the body). This further suggests that our model captures an underlying articulation prior.

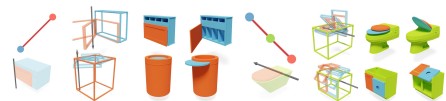

Figure 7: **Motion2Part**: Known motion structure (left) and suggested parts (right).

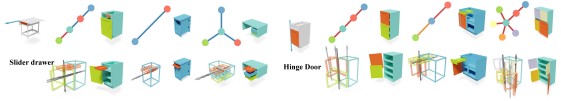

Figure 8: **GAPart2Object**: GAParts [81] (left) completed into full articulated objects (right).

**Motion2Part**. Similar to Part2Motion, we can flip the conditions and suggest possible parts from the motion structure. Given known edge attributes and graph topology $\mathbf{x}^{\text{known}} = \{e_{(i,j)}\} \bigcup \{o_i\}$ and $\mathbf{x}^{\text{unknown}} = \{T_{gi}, b_i, f_i\}$, NAP can generate possible parts that fit the motion structure in Fig. 7.

**GAPart2Object**. As Part2Motion and Motion2Part are pure edge or node conditions, we can also combine the node and edge conditions to complete an articulated object from one part. GAPart [81] is human-labeled semantically generalizable parts and one GAPart has a part geometry plus a joint type. We use the GAPart [81] from our testset as the known condition. Specifically, we set $\mathbf{x}^{\text{known}} = \{[o_0, b_0, f_0], [o_1]\} \bigcup \{[c_{(0,1)}, l_{(0,1)}, r_{(0,1)}]\}$ otherwise unknown, where we ensure that a GAPart is part-0 and there must be one part-1 connected to it. We constrain the joint direction $l_{(0,1)}$ but free the momentum $m_{(0,1)}$ and also let the non-zero joint range be freely denoised. The completion results are shown in Fig. 8.

## 5  Conclusions

We introduce the articulated object synthesis problem to our community and propose the first deep neural network that is able to generate articulated object models. Such generation is achieved via a diffusion model over a novel articulation graph parameterization with a graph denoising network. We further introduced a new Instantiation Distance to adopt the widely used 3D shape generation metrics to this task. Our learned neural 3D articulation can be used under different conditions for diverse application settings. We believe this initial step towards articulated object understanding and generation will bring more opportunities to understand and simulate our dynamic and complex real world.

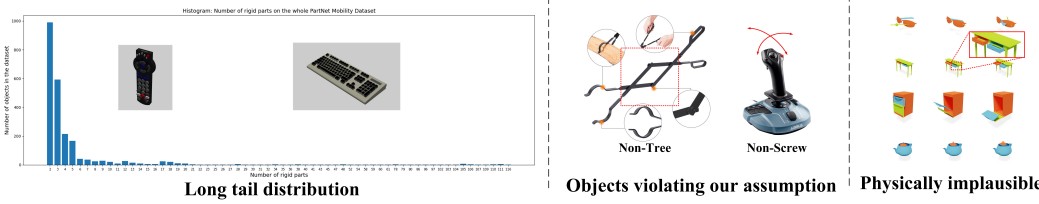

Figure 9: Limitations: **Left** The number of rigid parts in the current dataset has a long tail distribution, leading to extremely challenging objects, for example, a remote or a keyboard. **Middle**: Object violating tree or screw joint assumption. **Right**: The generated objects do not strictly obey physical constraints, for example, self-collision, making them not directly simulatable.

**Limitations and future work**. Although we have taken initial steps towards this new challenge, there are still many issues that require further exploration.

1. Long tail distribution and unbalanced structures: We noticed that our methods and baselines work better for small graphs. Large graphs with many parts are rare in the training set, making the generative model biased towards small graphs.

2. Overfitting: Like many other generative models, we do observe that NAP slightly overfits the training dataset. One bottleneck is that the current dataset still contains too few objects (around 2k) for training generative models. Studying how to learn an articulation prior from the camera observations or building a larger dataset will help to address this issue.

3. Tree and screw assumption: As shown in Fig. 9-Mid. NAP currently can not model kinematic cycles (pairs) or model several degrees of freedom on a single joint. Introducing dummy nodes and synchronization between joints may solve this limitation.

4. More structured diffusion: Although demonstrated to be effective, the diffusion we applied is straightforward and may be improved by introducing discretization, for example, the MST in the intermediate steps or by constraining the diffusion to the Plücker manifold.

5. Physically plausible generation: As shown in Fig. 9-Right, another exciting direction is how to generate directly simulatable objects whose joints and parts geometry fulfill physical constraints. This may require novel intersection loss and optimization techniques between parts and joints.

**Broader Impacts**. We do not see any direct negative ethical aspects or societal consequences of our work. Our paper can broadly improve the perception, digitization, and simulation of our physical world.

## Acknowledgments and Disclosure of Funding

The authors appreciate the support of the following grants: NSF FRR 2220868, NSF IIS-RI 2212433, NSF TRIPODS 1934960, NSF CPS 2038873 awarded to UPenn; TRI University 2.0 grant, ARL grant W911NF-21-2-0104, and a Vannevar Bush Faculty Fellowship awarded to Stanford University.

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
