# *NAP*: *N*eural 3D *A*rticulated Object *P*rior
## – Supplementary Material –

**Jiahui Lei[1]    Congyue Deng[2]    Bokui Shen[2]    Leonidas Guibas[2]    Kostas Daniilidis[13]**
[1] University of Pennsylvania    [2] Stanford University    [3] Archimedes, Athena RC
{leijh, kostas}@cis.upenn.edu, {congyue, willshen, guibas}@cs.stanford.edu

This supplementary document provides additional details of our method and baselines in Section S.1, as well as additional results in Section S.2.

## S.1    Implementation Details

### S.1.1    NAP and computation

**Shape Prior**. We pre-train an Auto-Encoder as in Occupancy Networks [1] (Fig. S1) to get the latent shape descriptors $s$ for the part shapes of all nodes. For each rigid part mesh, we preprocess it to a watertight mesh and sample the SDF for query positions in the normalized space. We also sample point clouds on the surface and input them to the shape Auto-Encoder. As shown in Fig. S1, the points are encoded to a global shape latent code with 128 channels by a Point-Net encoder, and then concatenated with the query positions and passed through an MLP to predict the SDF values. The reconstruction is supervised by the regression loss as in [2]. To train the shape prior, we use Adam optimizer with $lr = 0.0001$ and the learning rate has a step decay at step [100000, 150000, 200000] iterations with factor $0.3$. The batch size is set to 32.

**Training and Compute**. We train our denoising network following the standard scheme as in DDPM [3]. We use a batch size of 64 and randomly sample 10 diffusion time steps for each object in the mini-batch. The variance schedule $\beta$ follows a linear scheduling with total 1000 steps and $\beta_{\min} = 0.001$ and $\beta_{\max} = 0.02$. We train the diffusion model for $120k$ steps with an initial learning rate $0.001$ and decay at $40k, 70k, 90k$ with a factor of $0.3$. Since the diffusion process is defined on a compact articulation graph representation, the computation requirement of our model is lightweight and ours can run on limited computational resources. A typical training cost is 9 hours on a single RTX3090 GPU.

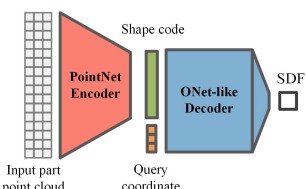

Figure S1: Shape Encoder-Decoder [1] for shape code $s$.

**Parameterization**. To facilitate the training, we count all the articulation tree attributes and normalize all channels to have a maximum value equal to 1 in the dataset. As mentioned in Sec.3.3 in the main paper, the input nodes are randomly permuted to break the permutation equivariance at the early reverse denoising stages. Note that there are two equivalent definitions of local joint coordinates, i.e., in the parent coordinate frame, flipping both the sign of the axes and the joint limits will result in the same joint. To eliminate such ambiguity, we canonicalize all the global axis Plücker coordinates to point toward the upper hemisphere (plus a half-plane and a line segment) and leave the sign to the chirality. When padding the tree to the complete graph, we leave the empty attributes to all zeros.

**Post-processing**. Upon completion of the entire diffusion generation process, we apply a final post-processing step to obtain the articulated object models (URDF). As we assume that we are modeling articulated objects, we ensure that there are at least two nodes in the final output articulation tree. This is achieved by applying a threshold of $0.5$ to node indicators $o$ to ascertain their existence. If

37th Conference on Neural Information Processing Systems (NeurIPS 2023).

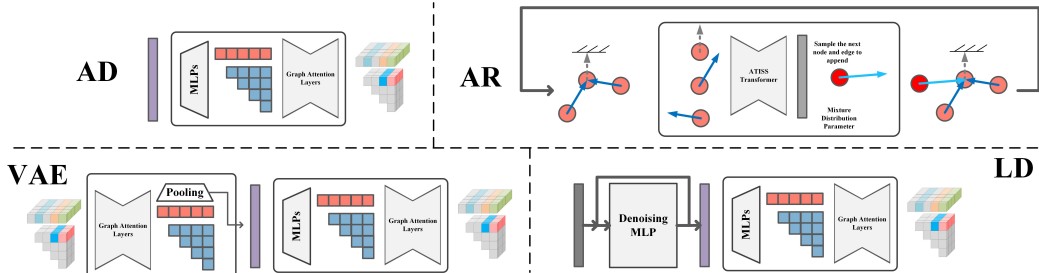

Figure S2: **Baselines**: AD stands for Auto-Decoding, AR corresponds to Autoregressive, VAE corresponds to Variational Autoencoding and LD is used for Latent Diffusion.

fewer than two nodes exist, we select nodes with the highest indicator values to compensate for the missing nodes. After the node identification, we extract an adjacency matrix of all possible edges between foreground nodes and use the chirality as the edge values. Since the denoising process is trained to predict high $|c|$ values on actual edges in an articulation tree and $|c| = 0$ for the padded dummy edges, we can use a minimum spanning tree algorithm to find the best tree structure from the predicted complete graph over $-|c|$. We also have to convert the predicted axis Plücker coordinates to a true unit Plücker as mentioned in our main paper Sec.3.1, 3.3 and limitations. We first normalize the predicted line direction $l$ to a unit vector and then subtract the parallel component of $m$ on $l$, ensuring that the final outputs $l, m$ are orthogonal. Finally, we can either apply an enhanced marching cubes algorithm [1] on the shape code descriptor predicted directly from the model to get the part meshes OR retrieve the nearest neighbors of the shape descriptors of the training set parts to get high-quality part meshes as most Scene synthesis [4, 5] do. The part meshes are scaled by the scale of the bounding box $b_i$ and translated according to $\boldsymbol{T}_{gi}$ in the node attributes.

### S.1.2 Baselines

We provide more details for the adaptation of the baselines to generating articulated objects (Fig. S2).

**Auto-Decoding (AD)**. One closely related task is the semantic-part-based shape generation, where the representative work is StructureNet [6]. Different from the semantic part hierarchy in [6], our task (rigid part) only has one level in the graph so it corresponds to one child-sub graph generation in [6]'s decoder ([6] Sec.5.2) and the edge in the graph has geometric meaning in our case. Since the StructureNet graph decoder is relatively small, for a fair comparison with our method, we adopt our graph-transformer to decode the articulation tree directly from a latent code as shown in Fig. S2-AD. A latent code is first decoded to the initial list of node and edge features, then processed by the same stack of graph attention layers as ours, and finally decoded to the articulation graph. We can directly learn the decoder with Auto-Decoding similar to [7] with regularization on the length of the latent vectors, which forms our first baseline.

**VAE**. We further adopt our graph-transformer to the StructureNet child graph encoder ( [6] Sec.5.1) to form an encoder and build a VAE as shown in Fig. S2-VAE. The input articulation graph is first processed by a stack of graph attention layers, and the node features, which already contain both the edge and node information after the encoding, are finally pooled to form a global latent code.

**Autoregressive (AR)**. 3D indoor scene synthesis is another task that is close to ours, with ATISS [4], an autoregressive method that iteratively generates objects in the scene, serving as a representative method. We adopt ATISS to generate articulation trees (Fig. S1-AR) by training the ATISS autoregressive model with the tree traversal order from random roots. The articulation tree is first parameterized to a list with each slot containing a node and the edge pointing to its parent, where the first node has a dummy edge pointing to empty. The order of the list is in a random tree traversal order, ensuring that the parent of each node must appear in the predecessor in the list. At each autoregressive step, the ATISS transformer aims to predict the mixture distribution parameters given the generated nodes and edges, so we can sample the next node and edge to append to the tree. Following ATISS, the termination of the above tree-growing iterations is also predicted by an indicator.

**Latent Diffusion (LD)**. Recent works [8, 9] have leveraged latent space diffusion in 3D shape generation. We adopt a similar approach to construct a baseline, as depicted in Fig. S2-LD Utilizing

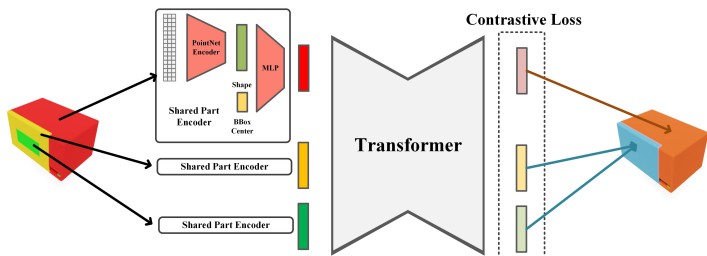

Figure S3: **Simple Part Grouper**: Given a part-over-segmented mesh, the network will predict for each fine-grained part a feature that can be used for grouping.

the trained decoder from the Auto-Decoding baseline, we learn a latent space diffusion model. This model generates the latent code used for decoding an articulation graph from Gaussian noise. The training and inference of the denoising network is similar to our method except the network now is a huge MLP with skip connections operating in the compact latent code space.

### S.1.3   Part Grouper

As mentioned in Sec.4.4-PartNet Imagination, we develop a straightforward simple part grouper (Fig. S3) to identify the rigid part since studying the distribution of hierarchy or structure inside rigid parts is not our focus. The input to the grouping network is the mesh of the finest leaf parts from PartNet [10], and the goal is to group these fine-grained parts into rigid parts by predicting for each node a feature that supports grouping. As shown in Fig. S3, the part shape is first encoded via sampling a point cloud and passing it through a PointNet to a shape code; then, the part position and bounding box are also encoded via positional encoding. The concatenation of these embeddings is encoded to the part feature by an MLP. Given a set of part features, a transformer further encodes the relationship between each fine-grained part and predict a feature for each. These features are supervised via a simple contrastive loss that increases the cosine similarity of the parts inside a rigid part group and decreases the similarity vice versa. Finally, a straightforward grouping method will identify the rigid group by taking a pre-defined threshold (we take $0.5$) and recognizing the rigid part groups by measuring their feature similarity compared to the threshold. Note that other (Part) instance segmentation methods should also work for this application.

## S.2   Additional ablations and results

### S.2.1   Ablation on $M$ for approximate Instantiation Distance

As introduced in Sec.4.1 in our main paper, the Instantiation Distance is approximated by uniformly sampling $M = 10$ joint angles or displacements in Eq.11. Note that computing the ID is expensive since it requires enumerating these $M$ joint states. One might argue that $M = 10$ could be insufficient. However, for computing the generative model metrics MMD, COV and 1-NNA, the dataset is large and the metrics are computed over the whole reference and sampling sets. As shown in Tab. S1, we compute the metrics for our method and the most competitive baseline Latent Diffusion (LD). The metrics between $M = 10$ and $M = 20$ are similar and we justify that $M = 10$ is a reasonable choice. Given that $M = 20$ samples provide more fine-grained sampling, the errors decrease accordingly

Table S1: Increasing $M$ when computing ID

|  | Part SDF Shape | | | | | | Part Retrieval Shape | | | | | |
|---|---|---|---|---|---|---|---|---|---|---|---|---|
|  | MMD↓ | | COV↑ | | 1-NNA↓ | | MMD↓ | | COV↑ | | 1-NNA↓ | |
|  | M=10 | M=20 | M=10 | M=20 | M=10 | M=20 | M=10 | M=20 | M=10 | M=20 | M=10 | M=20 |
| LD | 0.0314 | 0.0304 | 0.4365 | 0.4254 | 0.6269 | 0.6292 | 0.0288 | 0.0279 | 0.4477 | 0.4454 | 0.6102 | 0.6069 |
| NAP | 0.0268 | 0.0257 | 0.4944 | 0.4922 | 0.569 | 0.5679 | 0.0215 | 0.0204 | 0.5234 | 0.5412 | 0.5412 | 0.5445 |

### S.2.2 Ablation on different max numbers of parts $K$

We have also tested all methods with a greater predefined maximum number of parts, $K$, which is set to 8 in our main paper. The quantitative comparisons can be found in Tab. S1. It's important to note that the majority of the full dataset consists of less than 8 parts. However, as $K$ increases, both the training and testing datasets start to encompass slightly more objects with an increased number of rigid parts. As we have indicated in our limitations section, we observed a performance decline with the increase in $K$.

Table S2: Ablation: Predefined maximum number of parts $K$

| | Part Retrieval Shape | | | | | | | | |
|---|---|---|---|---|---|---|---|---|---|
| | MMD↓ | | | COV↑ | | | 1-NNA↓ | | |
| | K=8 | K=15 | K=20 | K=8 | K=15 | K=20 | K=8 | K=15 | K=20 |
| AD | 0.0390 | 0.0372 | 0.0351 | 0.2316 | 0.2004 | 0.1697 | 0.8675 | 0.8935 | 0.8828 |
| VAE | 0.0289 | 0.0293 | 0.0324 | 0.3363 | 0.2975 | 0.3071 | 0.7918 | 0.8070 | 0.8394 |
| AR | 0.0333 | 0.0335 | 0.0330 | 0.4120 | 0.4156 | 0.3960 | 0.6782 | 0.6677 | 0.6970 |
| LD | 0.0288 | 0.0284 | **0.0283** | 0.4477 | 0.4557 | 0.4465 | 0.6102 | 0.6266 | 0.6687 |
| NAP | **0.0215** | **0.0274** | 0.0288 | **0.5234** | **0.4810** | **0.4889** | **0.5412** | **0.5696** | **0.5697** |

### S.2.3 Human Evaluations

We further evaluate our generation results using a user study following [11]. In each questionnaire question, users are provided 5 groups of generated 3D models with either our method or the baselines and asked to rank the results produced by 5 models according to realism and physical plausibility. The average rankings are shown in Tab. S3. Our method obtains the highest rank by the users compared to the baselines.

Table S3: User study results

| Method | AD | VAE | AR | LD | **Ours** |
|---|---|---|---|---|---|
| Avg. rank | 4.44 | 3.37 | 3.63 | 3.00 | **1.24** |

### S.2.4 Graph Statistics

We show statistics on the number of nodes and degree of nodes in the articulation tree in Fig. S4. Our generated distribution matches the ones from the ground truth training and testing set.

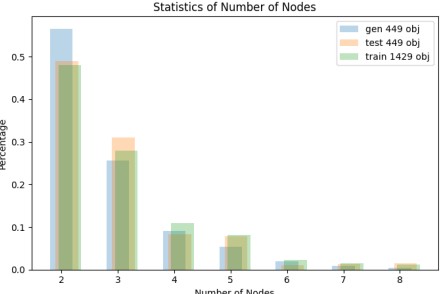 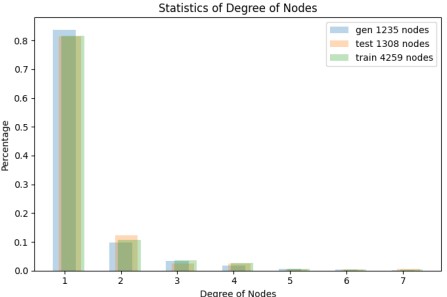

Figure S4: **Statistics on Graphs**: the generated tree topology is shown on the left, where the edge color means blue – prismatic, red – revolute, and orange – hybrid; The generated bounding boxes plus joints and the retrieved part shapes are shown on the right. The possible joint angles are visualized from the min to the max.

### S.2.5   More visualizations

We show more visual results for our unconditional generation in Fig. S5 and for our applications: Part2Motion in Fig. S6, Motion2Part in Fig. S7, PartNet Imagination in Fig. S8 and GAPart2Object in Fig. S9.

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

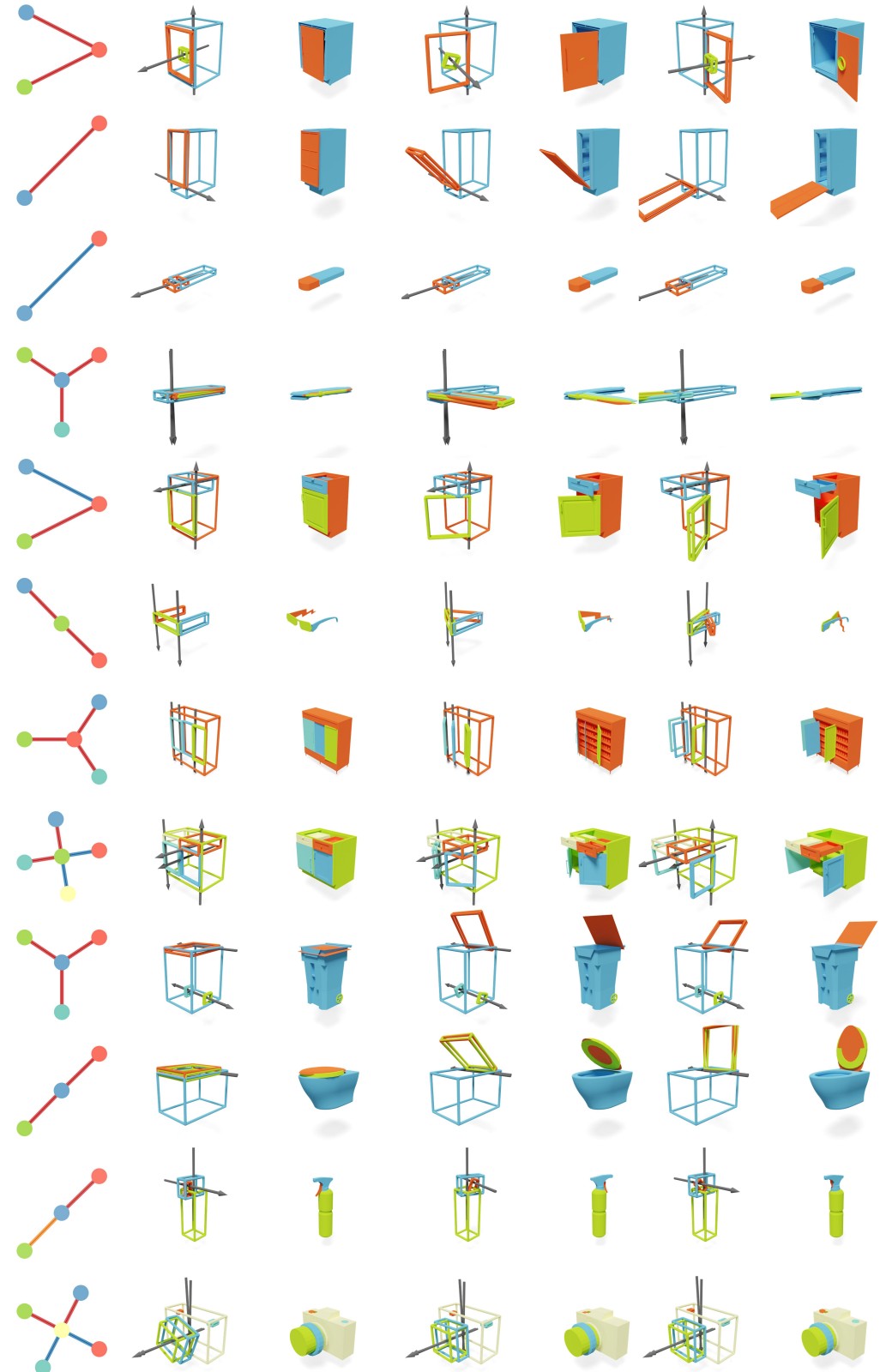

Figure S5: **Unconditional Generation**: the generated tree topology is shown on the left, where the edge color means blue – prismatic, red – revolute, and orange – hybrid; The generated bounding boxes plus joints and the retrieved part shapes are shown on the right. The possible joint angles are visualized from the min to the max.

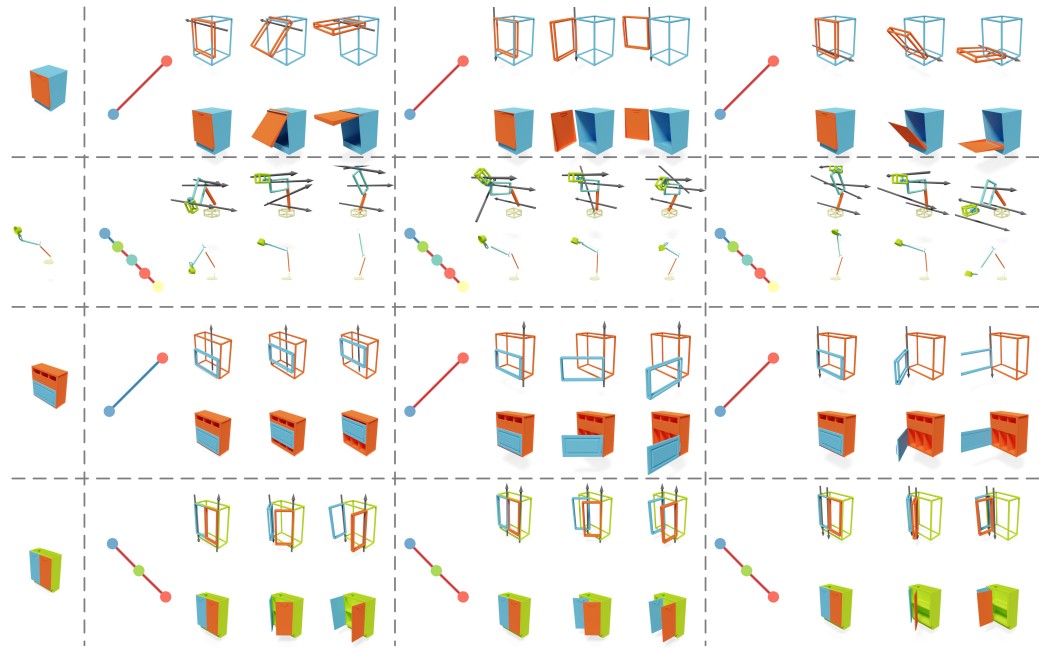

Figure S6: **Part2Motion**: The input node conditions are shown on the left and the diverse completion results are shown on the right in a similar format with Fig. S5.

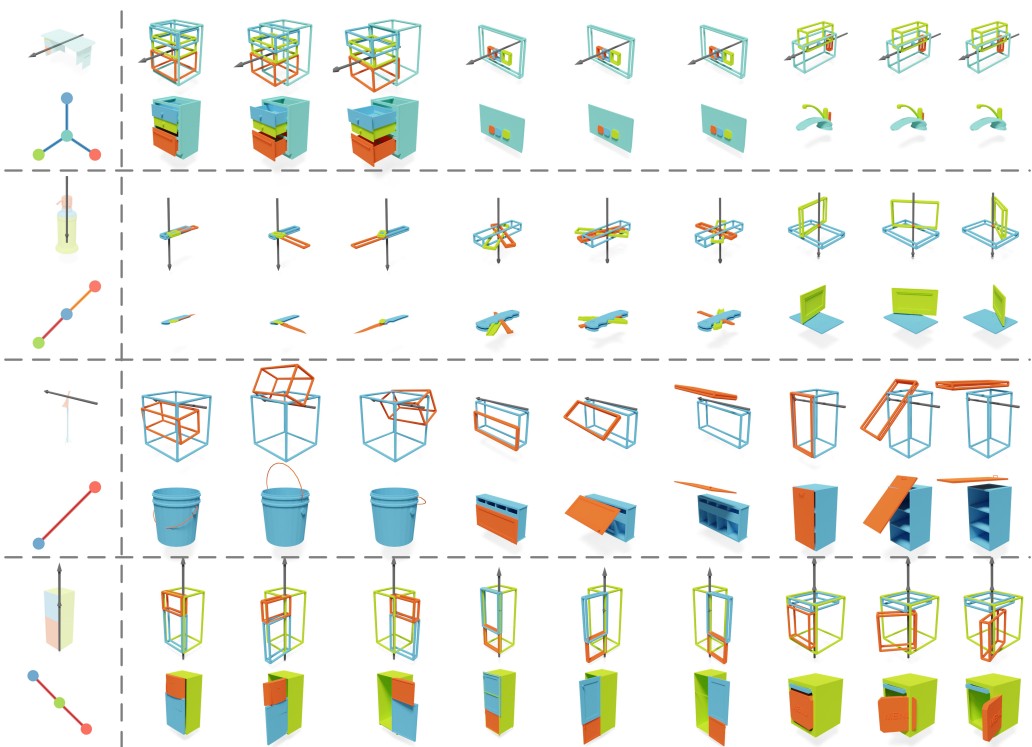

Figure S7: **Motion2Part**: Similar to Fig. S6, the motion structure conditions are shown on the left and the generation results are on the right.

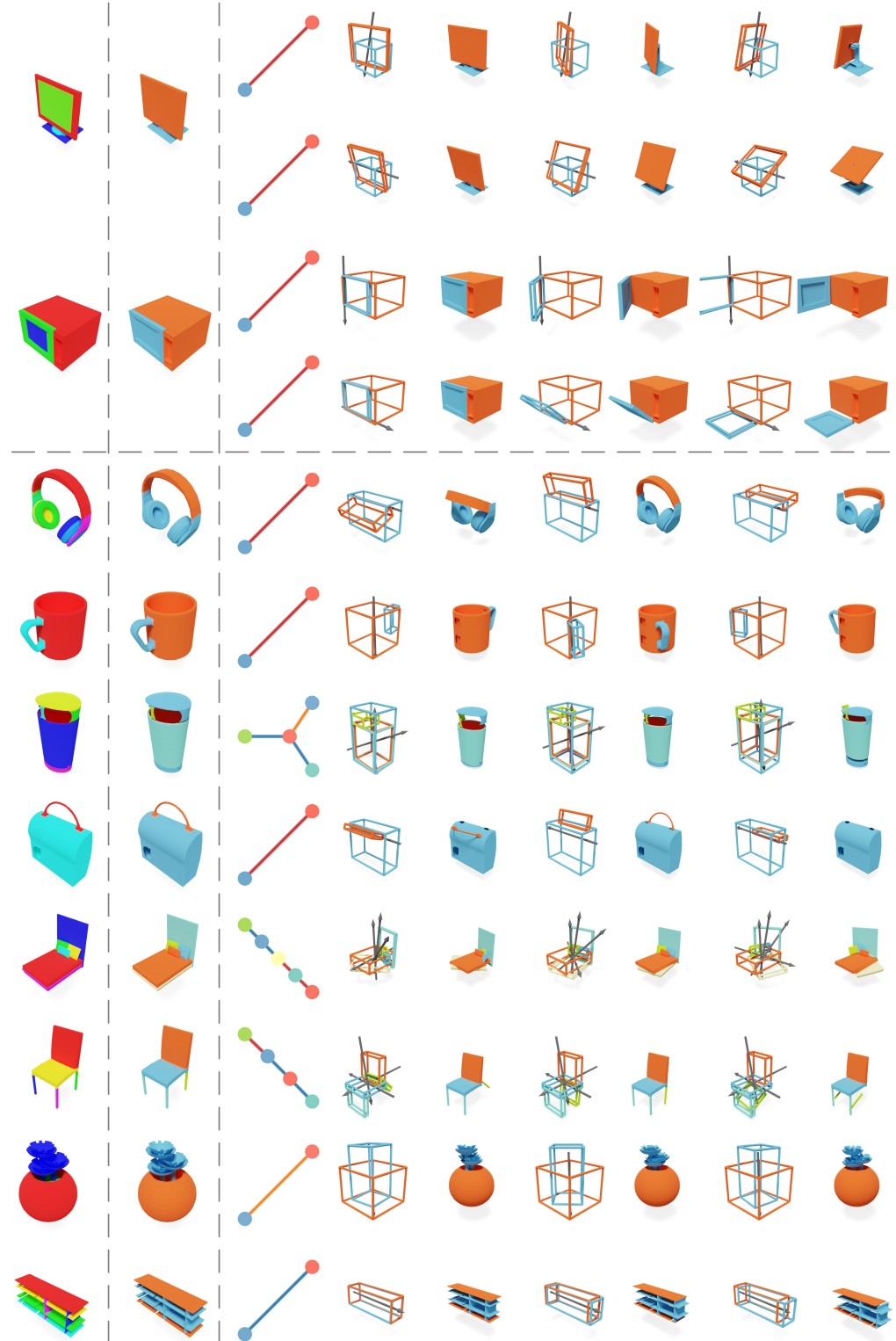

Figure S8: **PartNet Imagination**: The input fine-grained over-segmented static PartNent [10] objects are shown on the left; the grouped parts are shown in the second column; and the generated articulation objects are on the right. The top two rows show the in-distribution objects with diverse motion proposals and the rest shows out-of-distribution objects in a similar format with Fig. S5

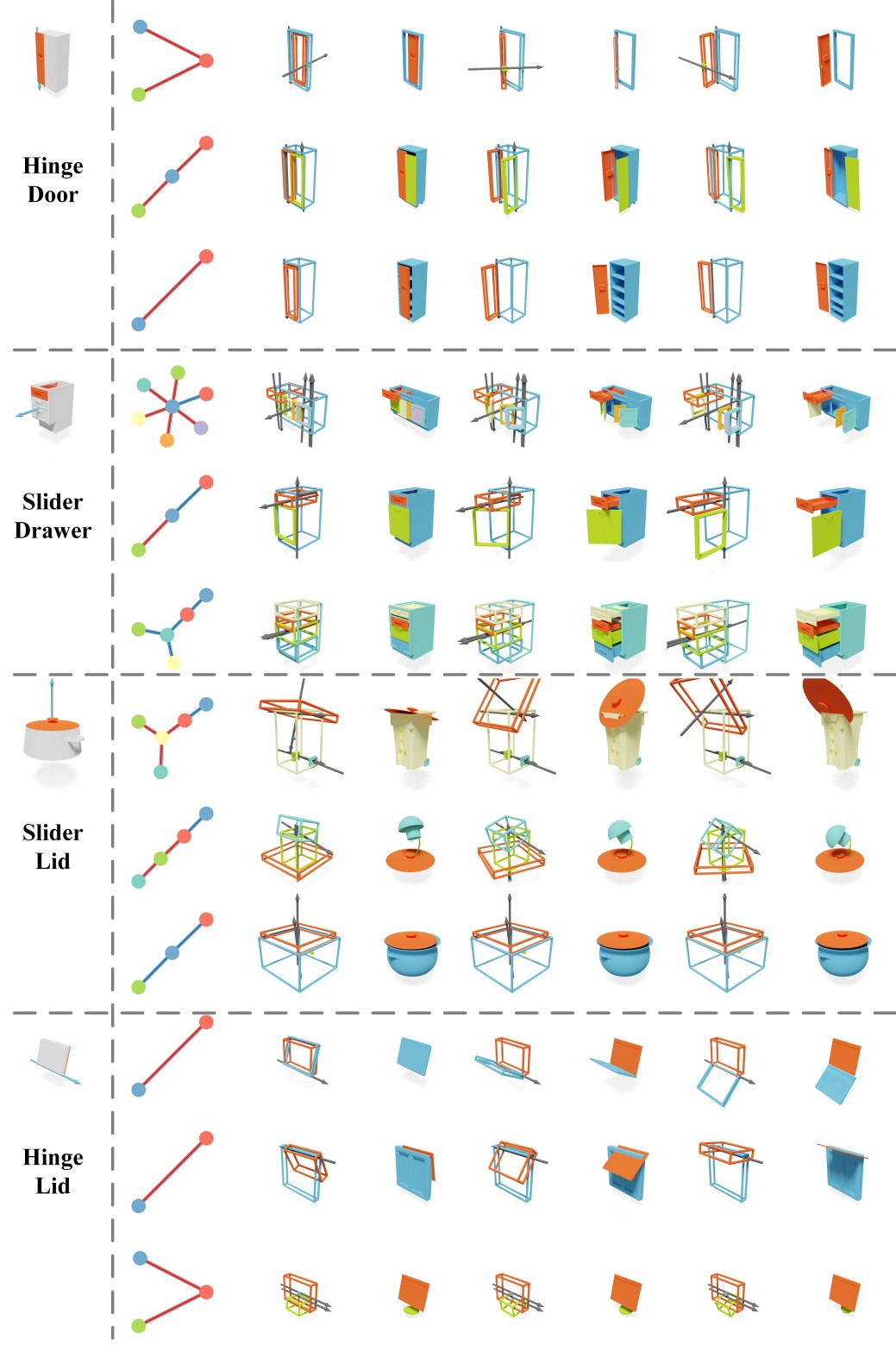

Figure S9: **GAPart2Object**: The GAPart [12] conditions are shown on the left and the generated objects are shown on the right in a similar format with Fig. S5.