# OpenReview forum: "NAP: Neural 3D Articulated Object Prior"
_NeurIPS.cc/2023/Conference — NeurIPS 2023 poster_

### Official Review · Reviewer_WyGQ · 2023-07-04

**Soundness:** 3 good
**Presentation:** 1 poor
**Contribution:** 3 good
**Rating:** 7
**Confidence:** 4

**Summary:**

This paper introduces a diffusion-based generative approach targeting daily articulated objects as a novel target category. The model outputs part shapes and joint configurations based on the proposed graph representation. The paper also proposes a novel transformer network to accommodate the graph structure in the generative process. The proposed method is evaluated on the Part-Mobility dataset using the newly proposed evaluation metric for the task, and it demonstrates superior performance over self-made baselines of non-diffusion-based and diffusion-based generative model architectures. The paper also demonstrates several applications of conditional generation on seen and unseen synthetic datasets.

**Strengths:**

* The paper tackles the previously unaddressed setting of generating daily articulated objects and the proposed approach seems promising.
* The approach demonstrates superior generative performance over the baselines both qualitatively and quantitatively.
* The paper demonstrates novel conditional generative applications and demonstrates generalizability to unseen datasets.

**Weaknesses:**

# Major
Although I found this work exciting and potentially interesting to the related audience, I feel the current writing quality is not sufficient for a conference paper.

* Inconsistent/confusing/unexplained notations
  * L111,116: The bold font of T_{gi} is inconsistent.
  * L118,216: The italic font of SE is inconsistent.
  * L113,116, and supp. L9: The implicit shape latent code s_i is written as f_i or f, which is used as a node feature in L194.
  * In L109, L129: The transformation from local to global coordinate is implicitly expressed as switched subscript: _{ig} and_{gi}. But it’s hard to understand at first glance.
  * L109,217: "i-th" and "ith" are inconsistent.
  * L215: The definition of T_{part} is missing.
* Missing reference to supp.: The implementation details of the Graph layer are missing in the supp., although it’s referred to in L199. How global pooling is applied is unclear.

# Minor
The training detail of the pre-trained shape prior network is missing in the supp.

**Questions:**

# Questions
* How is either prismatic or revolute enforced? In L125, how is a node’s joint type decided as prismatic, revolute, or hybrid using r(i,j)? Is there some threshold?
* Why not use an indicator variable to determine either prismatic or revolute?
* In S.1.1, how do you reflect node existence o in MST over chirality? Do you make the chirality of the edges having the node zero?

# Suggestions
I suggest adding the following visualizations for a better understanding of the paper:
* Qualitative visualization of the ablation
* Visualization of failure cases
* Schematic visualization of the variables described in Sec. 3.1.

**Limitations:**

Limitations are explained in the conclusion of the main paper. However, visualizations of some of the limitations and failure cases are missing, such as physically implausible generation. Adding those visualizations would help further understand the limitations.

---

> ### Author Rebuttal · Authors · 2023-08-08
>
> Thanks for your feedback! Here are our responses to your questions and comments and we hope that they could help to address your concerns:
>
> - **Writting and schematic visualization**: We really appreciate these careful checks and suggestions and we take all these seriously. Please see [G1] in the global response as well. As suggested, we append a draft of the schematic visualization in the attached Fig.R8.
>
>   In L215, $T_{part}$ represents the part pose in the global object frame. We will correct all the typos, inconsistent notations, and confusing subscripts as mentioned, and complete the missing details in our revision.
>
> - **Graph layer and shape prior training details**: We apologize that some graph layer details are missing in Suppl. Sec.S.1: The global pooling in the graph layer is similar to a point-net – doing a max pooling over all node features and concatenating this global pooled feature back to each node again.
>
>   Regarding the shape prior training, besides L8 in Suppl. here are more details: the optimizer is a standard Adam with lr=0.0001 and the learning rate has a step decay at [100000, 150000, 200000] iterations with factor 0.3. The batch size is 32 and we use the model checkpoint at epoch 737. We will add all these details to the supplementary, and the code that contains all implementation details will also be released once the paper is published.
>
> - **Joint type**:  Leveraging the screw representation [56], as in L126, if a joint is revolute, its ground truth prismatic working range will be [0,0] and vice versa. This is softly enforced by the loss that learns to predict [0,0] through denoising. To decide the joint type for visualization purposes (in Fig.4 left top corner), yes, there is a threshold to test whether the working range is large enough so the corresponding mode really exists. We use th=0.003 in the paper visualization for both prismatic and revolute modes. “Why not use an indicator for these?” We agree that this is a valid design but it would potentially add two additional variables that need to be predicted, so we directly decide the joint type from the joint range prediction, fully exploiting the simplicity in the screw representation.
> - **Existence of nodes:** Sorry for this confusion. There are two “indicators” in our representations, The variable o serves as the node existence indicator, while c indicates edge existence. During the final output extraction (as described in L163 and Suppl. L39), we first identify node existence by thresholding the node indicator o at 0.5. All existing nodes at this point form a complete graph (a sub-graph of the original padded one). We then utilize the chirality absolute value −∣c∣ of each edge to perform the MST on this sub-complete graph, determining which edges exist.
> - **Visualization of ablation and failure cases:** We append the visualization of ablation in the attached Fig.R6 and of failure cases in Fig.R7. Note that in Fig.R7 highlighted area, the drawer is not physically plausible, which means it can not be simulated with self-collision in a physical simulator, we hope future work will tackle this problem. We’ll append these figures to our revision.

---

> > ### Comment · Reviewer_WyGQ · 2023-08-17
> >
> > I thank the authors for their effort to address my questions. I have no further questions or comments at this time.

---

> > > ### Author Response · Authors · 2023-08-17
> > > **Thank you!**
> > >
> > > Thanks for your feedback and we really appreciate your time!

---

> > ### Comment · Reviewer_WyGQ · 2023-08-21
> >
> > Once again I thank the authors for the detailed responses. After I carefully read all the reviewers' comments and the corresponding authors’ responses, I would like to raise my score to accept. The reasons are as follows.
> > The authors addressed my suggestion in the attached figures. The visualization of the failure cases and schematic visualization of the variables are satisfactory and would help better understand this paper for the readers.
> > In writing, considering the response both to my review and the reviewer ncuk, the authors added a detailed response to the listed writing errors/questions + the draft. I would expect the authors carefully revise the manuscript before submitting camera-ready if accepted.
> > Therefore, I believe the technical merits of this paper outweigh the remaining concern about the limited size of the dataset.

---

### Official Review · Reviewer_hNRt · 2023-07-05

**Soundness:** 3 good
**Presentation:** 3 good
**Contribution:** 2 fair
**Rating:** 6
**Confidence:** 4

**Summary:**

The paper presents the task of generating articulated objects, encompassing the generation of both structurally and geometrically plausible objects. To address this task, the authors propose a novel articulation "complete-graph" parameterization. This parameterization encodes the geometry and part poses within the nodes, while representing the joint constraints through the edges. To implement the generation process, the authors employ a diffusion model with their designed graph fusion module. The paper also introduces a novel distance metric for evaluating the distribution of the generated articulated objects. Additionally, the paper delves into various applications leveraging conditioned generation.

**Strengths:**

* The proposed task to generate articulated objects is interesting and the motivation to learn the distribution of the articulated objects is intuitive.
* The “complete graph” representation for the articulated objects is effective to decompose the geometry and structures into the nodes and edges of the graph. Through setting a max number of nodes K, the parameterization can successfully encode most of the articulated objects into a consistent complete graph representation for the following generation process.
* The graph layer to fuse and update the information in the node and edges is helpful based on the results of the ablation studies.

**Weaknesses:**

* For a generation task, it’s hard to evaluate the novelty of the generated objects with existing evaluation metric mentioned in the paper or the user study to compare the results from different baselines. The dataset used in this paper is PartNet-Mobility only contains 2346 objects, after filtering objects with more than 8 parts, there are about 2000 objects. Based on the split mentioned in the paper, there will only be about 1400 articulated objects in the train set used to train the diffusion model. It’s highly easy for the diffusion model to overfit on such a small set of articulated objects. Some potential way to evaluate the novelty and uniqueness of the generated model is to fetch the most similar objects in the train set and fetch the most similar objects in the generated set to qualitatively see the difference. Current evaluation and qualitative results cannot show if the generated models are a copy of the models in the train set.
* In this paper, actually the task simplify the general articulated object generation task, which only generate articulated objects whose all parts are in the rest state. The authors also assume the initial state of the articulated models in the PartNet-Mobility dataset is close state, which doesn’t hold true for all models in the dataset. For example, some cabinets in the PartNet-Mobility dataset have some door open, while some are close as the initial state.
* The generation task of the articulated objects is demonstrated to focus on both geometry and the motion structure. However, for the part geometry, the authors choose to retrieve the most similar parts in the latent space for some visualization and potentially for the evaluation. In this way, actually the generative task focuses on generate the combinations of the parts instead of really generate the articulated models.
* The direction of the edge is explained in the parameterization, however, not used when constructing the minimum-spanning tree. It’s unclear how the edge chirality (+1, -1) are used.

**Questions:**

* For the evaluation, the part geometry uses the retrieved ones or directly the results from the occupancy network? Because from the rendered results used for evaluation in the supplemental material, it seems that the part geometry is more likely the retrieval results.
* For the conditioned generation, it’s easier to compare the results with some existing work. For example, for the part2motion, is it possible to quantitatively evaluate the generated results? How many generated models will cover the GT motion? And based on the results from Part2Motion, it seems that the handle hints in the part are not fully understood by the network. Are there some results to show that the NAP can really understand the joint constraints based on the part geometry?
* For the generated structures, is there some statistics on the number of nodes and number of edges of the generated set and the train/test set?
* For the parameterization of the articulated objects, the order of the node seems also important to be somehow consistent. Are there some operations for the node order when parameterzing the articulated objects (e.g. part from top to down)

**Limitations:**

The authors mention the limitation.

---

> ### Author Rebuttal · Authors · 2023-08-08
>
> Thanks for your feedback! Here are our responses to your questions and comments and we hope that they could help to address your concerns:
>
> - **Overfitting?**: Please see response G2 in the global block.
>
> - **Rest state**: We appreciate the reviewer’s careful observation of PartNet-Mobility. We will explain this more clearly in Sec 3.1:
>
>   (1.) NAP does not require a strict restriction of what is “rest”: NAP chooses to define the parts and joints in a global object coordinate system for simplicity and any well-defined articulated object can be instantiated and represented in such an object frame. The object articulation pose does not have to be what human thinks is “rest”.
>
>   (2.) A relatively “canonical” rest state is a reasonable situation in the current existing datasets, from which NAP can take advantage: As articulated object annotation is expensive, most of the objects have some consistent state presented when labeling the dataset. For example, PartNet-Mobility is mostly from PartNet which is eventually from ShapeNet. No matter whether the door is closed or open, there does relatively exist global consistency in the dataset that forms relative canonical states (even if half of the doors are open and half of the doors are closed, there is a two-mode canonical state), from which NAP can take advantage and learn a more canonical and stable prior.
>
> - **Retrieval or reconstruction?** We will emphasize this more obviously in our revision.
>
>   (1.) We evaluate with both the implicit part reconstruction and the retrieval in Tab.1 and Tab.2 (left is recon and right is retrieval). Fig.4 (the third column of each object viz group) in our main paper also visualizes both cases.
>
>   (2.) Our primary emphasis is on the generation of structured objects. Like many scene synthesis methods [19-30], we retrieve the nearest part shape from the training set to enhance quality. We anticipate that future works can further refine the quality of individual parts using advanced shape-generation techniques.
>
> - **Chirality:** Sorry for this confusion, and we will provide a clearer explanation in our revision:
>
>   (1.) In L132, we highlight the simplicity of global plucker coordinates (against defining in local parent or child coordinates), i.e. $p_{(i,j)} = -p_{(j,i)}$ and $r_{(i,j)}=r_{(j,i)}$, which motivate us to only represent compactly $K(K-1)/2$ edges as in L133. But note that the nodes have no specific order (randomly permuted) so that the edge between node i and j in these $K(K-1)/2$ may sometimes has direction $(i,j)$ but sometimes $(j,i)$. For a more elegant representation, we explicitly model the negative sign of $p_{(i,j)} = -p_{(j,i)}$ into chirality as $+1,-1$ and let the representation always have one consistent plucker $p$ for the edge between node i and j ignoring the edge direction.
>
>   (2.) In practice, after applying the MST based on negative chirality $-|c_{(i,j)}|$, we determine the direction of each existing edge: if $c_{(i,j)}>0$ we do nothing; if $c_{(i,j)}<0$ we multiply the predicted plucker coordinates with $-1$, which is equivalent to claiming that this predicted joint parameter is actually with direction $(j,i)$ instead of $(i,j)$.
>
> - **Part2Motion**: Our primary focus in this first step towards deep articulated object generation is unconditional synthesis, and the applications further demonstrate our prior is useful and flexible.
>
>   (1.) As requested, in Part2Motion, we count how frequently the ground truth joint axes are covered by the generated objects (within 5 degrees, 0.05 distance, 20 generations per GT object) and the results are reported in Fig.R2. On the testing set, we observe an average of 38.64% probability that the ground truth joint will be covered by the conditional generated joints. As these application tasks are quite new and their evaluation and task definition are non-trivial, we leave them for future work in the area to further explore.
>
>   (2.) Indeed, we found that the small handle hints are not fully parsed in Fig.5 in our main paper: the generated motion includes joints near the handle side. This is a limitation of our current method. One potential reason is that the simple pretrained shape Encoder-Decoder (PointNet+OccNet) is not detailed enough and the learned part shape latent space is not sensitive enough to these small geometry details. We hope this would be improved by future approaches and we will append this to our limitation, thanks!
>
> - **Statistics on Nodes and Edges**: These are usually reported when studying the generation of general graphs[83]. We report the statistics of the number of nodes and node degrees (reflecting some structure) in attached Fig.R3. Note that since the generation always produces a tree, there is no need to count the number of edges. We observe that NAP can generate close distribution to the training and test set.
> - **Order of nodes**: We don’t have any specific spatial order of the node in the parameterization. However, in order to make the network more expressive, especially during the early denoising stages, we do add the positional encoding of the order (the #th position in the list) to the network (inspired by [22]), and during training, we randomly permute the node order in the graph.

---

> > ### Comment · Reviewer_hNRt · 2023-08-18
> >
> > Thanks for the responses from the authors. Although I still have the concern on limited data size for articulated models and the usefulness for such generation task, I think this paper can motivate more papers to follow up on this direction. The learned distribution definitely can support a variety of downstream tasks. I hope the author can address the limitation of current work more clearly in the final version. I will raise my score.

---

> > > ### Author Response · Authors · 2023-08-19
> > > **Thank you!!**
> > >
> > > Thanks for your feedback and we really appreciate your time!
> > >
> > > Definitely, we will improve the presentation and address more clearly the current limitation in our final version, thanks!

---

### Official Review · Reviewer_17WX · 2023-07-07

**Soundness:** 4 excellent
**Presentation:** 3 good
**Contribution:** 4 excellent
**Rating:** 8
**Confidence:** 5

**Summary:**

This paper proposes Neural 3D Articulation Prior (NAP) to synthesize 3D articulated object models. The key contributions include (a) an articulation tree parameterization for the diffusion denoising probabilistic model and (b) a new distance function for evaluation. The paper also shows quantitative and qualitative improvements over prior methods.

**Strengths:**

1. This paper introduces a new problem of articulated object synthesis. While there are some prior works for (mostly) category-specific articulated object modeling, this paper focuses on generative modeling across different categories.

2. There are a few key innovative components in the proposed methods such as the articulation tree parameterization and graph-attention denoising network. Experiments (Tab. 2) have also shown the effectiveness of these new components.

3. This paper has also proposed new metrics (distance) to evaluate this new task, which may benefit future works for a fair and efficient comparison.

4. Implementation details are well-documented.

**Weaknesses:**

1. A neural implicit surface is used to fit each part. From figures such as Fig. 4, it seems the geometry of the proposed method is not very high-quality. However, it seems the geometry of single parts is simpler than the geometry of the entire shape, and prior work (eg, [100, 110]) has shown much better geometric results even for an entire object. Could the authors explain why the synthesized neural part surface looks worse than synthesized neural single-object surface from prior work?

2. The entire pipeline and the key components all seem very valuable to the community. To help us better understand what is easier for the network to learn, could the authors share some examples of failure cases / more challenging types or categories?

**Questions:**

One suggestion is to improve Fig. 3. Currently, it seems too packed and the text is too small to be read.

**Limitations:**

The authors have adequately addressed the limitations and broader impacts.

---

> ### Author Rebuttal · Authors · 2023-08-08
>
> Thanks for your feedback! Here are our responses to your questions and comments and we hope that they could help to address your concerns:
>
> - **Part shape quality**: We appreciate this insightful observation. Here are some potential explanations:
>
>   (1.) **Focus on articulated object generation**: Many scene synthesis methods, similar to ours, don't primarily focus on individual object shapes but rather retrieve them from a database. As our main emphasis is on generating structured objects instead of single-part shapes, we take the most naive way to model the part shape, an Auto-Encoder, even with NO regularization in the latent space. Here we do an additional experiment: when learning the pre-trained part shape module, we add regularization to the latent space (VAE loss) and reduce the latent dim from 128 to 64, aiming to cultivate a more meaningful part shape latent space. After learning NAP on this updated part shape prior, we do observe the part quality is improved a little, and the quantitative results evaluated with the predicted part SDF geometry improve as shown in the table:
>
>     |                            | MMD ↓      | COV ↑      | 1-NNA ↓    |
>     |----------------------------|------------|------------|------------|
>     | Paper                      |     0.0268 |     0.4944 |     0.5690 |
>     | With VAE regularized prior | **0.0229** | **0.5167** | **0.5490** |
>
>   We believe that leveraging advanced part representations, like tri-planes or local feature grids, can further enhance reconstruction quality. We will highlight this in our limitation section.
>
>   (2.) **Thin structures**: Unlike shapes in ShapeNet, the rigid parts of an object usually exhibit very thin structures. In the PartNet(Mobility), a lot of parts are non-watertight and have large areas of single side surface: for example, a door may be just one plane, with zero thickness. To learn an SDF, we apply heavy pre-processing on these ill-defined thin structures. We know that vanilla DeepSDF or OccNet struggle with such thin structures (e.g. the airplane wings), which might be the biggest reason for the quality drop. Potentially, more expressive local methods or unsigned distance functions may help to solve these problems.
>
>   (3.) **Part shape variance**: We are modeling rigid parts, whose shape variance may not be as small as we first thought. For example, the network has to learn very simple shapes – a door, a round button, or a drawer that may appear repeatedly in the dataset, and simultaneously learn relatively complex shapes – a whole chair seat back plus arms and the bottom, or the whole body plus base of the storage furniture that are relatively rare in the dataset. What’s worse, as the rigid part in our dataset has no semantic labels or the concept of categories, it’s not easy to balance the part samples during training between these simple and hard shapes. As a result, the network may be biased to small and simple repeating parts but have difficulty capturing more complex ones. This issue may be resolved by a smarter training strategy or network design for the part shape prior.
>
>   (4.) **Prediction**: as the shape latent code is predicted from the denoising, we can not guarantee that the accuracy in the latent space is perfect. As small changes in the latent space may lead to un-plausible changes in the output mesh, suggesting room for improvement in this task.
>
> - **Failure cases and more challenging types or categories**: We've included visualizations of failure cases in the attached Fig.R7. Fig.R5 showcases examples that currently violate our assumptions, which might be addressed with dummy nodes. Fig.R4 highlights the long-tail distribution of articulated objects in our dataset, with items like remotes and keyboards being particularly challenging. We'll incorporate these insights into our revision.
> - **Figure suggestion**: Thanks for this suggestion. Due to page constraints, the figure was downsized. We'll restructure the figure layout in our revision for better clarity.

---

> > ### Comment · Reviewer_17WX · 2023-08-17
> >
> > The authors have answered my questions. After reading all reviewers' comments and the responses from the authors, I am leaning towards keeping my original rating.

---

> > > ### Author Response · Authors · 2023-08-17
> > > **Thank you!!**
> > >
> > > Thanks for your feedback and we really appreciate your time!

---

### Official Review · Reviewer_ncuk · 2023-07-09

**Soundness:** 3 good
**Presentation:** 2 fair
**Contribution:** 3 good
**Rating:** 4
**Confidence:** 2

**Summary:**

This paper proposes a diffusion model based 3D generative model for articulated objects. The major contribution is the 1) tree representation of the articulated objects, 2) the corresponding graph-based diffusion models, and 3) a distance metric for evaluation. The proposed framework works well on PartNet-Mobility objects, beating baselines based on other generative models, and enables a series of conditional generation applications.

**Strengths:**

- The motivation is clear. To catch up the latest trends and use the diffusion models, the tree/graph parameterization is developed and implemented in a neat way.
- The proposed solution works well on the tested dataset and outperforms other generative baselines including a latent diffusion model.
- The downstream applications are interesting and demonstrates one important advantage of this framework: conditional generation is easy.
- The graph attention denoiser is reasonable and serves well in the entire framework. Usually graph-based network doesn't generalize very well.
- The proposed distance metric makes lots of sense.

**Weaknesses:**

### Major: (my current rating is based on my concerns as listed below)
- Doing diffusion process on a complete graph is computationally heavy and slow. I'm not sure how well this method scale. The illustrated graph are all relatively easy with few nodes and edges. Please see below section for my detailed question regarding this point.
- The writing is unclear and I quickly lost in the writing, especially in the approach section. Some of the issues can still be figured out through context, but others really hurts the readability (eg., I had a hard time understanding the edges section). Please see my detailed comments in Question section. I try to list as many as I can but probably still miss a lot.
- In many places, this paper claims they are "the first 3D deep generative model to synthesize 3D articulated objects" or "introducing the articulated object synthesis problem". This is a bit over-claimed since 3D articulated object is a very broad domain, which also includes animals, human, and many other things. There is big literature on 3D human/animal generation in computer vision field. Meanwhile, the motion patterns and the graph-scale studied in this work probably only works for robots and simple objects. Therefore, it's better to tone down a bit.
- Is PartNet-Mobility the only dataset can be used? The objects in this dataset are relatively simple in geometry and motion. Is it possible to generalize to more diverse and realistic dataset containing more articulated objects like animals/human or more challenging objects?

### Misc:
- In the official Formatting Instructions Line101-102, it stats that "The table number and title always appear before the table". The submitted paper doesn't follow this.
- Approaches [3.1]: Why using Plücker coordinate to represent joints? Is this the only option? If not, what's the advantage of using this over the other alternatives? The motivation is a bit unclear.

**Questions:**

There are some unclear writings which can be clarified:
- L109: is the initial pose $T_{gi}$ a node property? This is different to the statement in L106 "Every joint (edges) has an initial pose"
- L112: why a 3D bbox $b_i$ is in $\mathbb{R}^3$? Three numbers represents a point in 3D space.
- L115: $o_i$ is a **per-part** binary indicator of part existence. If there are K parts, I think $o_i$ should be in dimension {0,1}$^K$?
- L116: $f_i$ is a new notation and not introduced before. I assume this is a typo, which should be $s_i$?
- L121: the domain of $l$ is $\mathbb{S}^2$: what is $\mathbb{S}$?
- L124: "... to avoid local coordinate changes caused by parent-child order flips" -- > when will parent-child order flips happen? How bad is this problem in practice?
- L125: for two ranges $r$ domain $\mathbb{R}^{2\times2}$: can the lower range be negative values?
- L128: joint states $(\theta, d)$ is not defined. What are the meanings of these two variable?
- L128: the relative transformations $T(\theta, d)$ is between two parts, let's say i and j. Then is the joint state $(\theta, d)$ of i or j?
- L129: global joint axis $(l_g, m_g)$ is undefined before using. How are this global axises defined?
- L129: $R_{ig}$ and $t_{ig}$ are undefined before using. The authors might want to state in L109 that $T_{gi} = [R_{gi};t_{gi}]$. In addition, the subscripts are not consistent.
- L136: based on the description in L121, $l\in\mathbb{S}^2$ and $m\in\mathbb{R}^3$. Then $p_{i,j}$ should be of dimension 5 not 6?
- L136: $c$ is a discrete value and hence not in $\mathbb{R}$.
- L146: I find this domain definition strange. v and e are different variable and you cannot simply adding their dimension as the graph dimension.
- L155/Eq.4: $\theta$ is already defined in Eq1. Please change variable name.
- Occupancy Networks predicts occupancy score not SDF value. In many places of this work, SDF is mentioned as OccNet output. Just want to clarify, which one is used?

Other more conceptual questions:
- Following my point regarding memory consumption in weakness section, what's the average graph size and the corresponding memory usage and training time for current framework? What's the biggest graph size the current solution can afford on a standard GPU during inference time and what's the corresponding memory usage? The illustrated graphs in this work are all relatively easy with few nodes and edges (L240 maximum 8 rigid parts). I saw that the limitation section first point mentioned this, but please be more detailed.
- It's unclear to me why doing a graph diffusion on a complete graph is better than doing in latent space. What advantages do we gain for using this graph diffusion process? On the other hand, the disadvantages are obvious: high memory consumption and tricky graph-network design.


**Limitations:**

The authors discussed the limitations in L311-318. I agree with these points in general but I do think one important point is missing. The approach is based on two assumptions: Tree assumption and Screw joints. The limitation of these two assumptions should be mentioned. For what kind of objects will they break?

I agree that the potential negative societal impact isn't an concern for this work as mentioned in L325.

---

> ### Author Rebuttal · Authors · 2023-08-06
>
> Thanks for your feedback! Here are our responses to your questions and comments and we hope that they could help to address your concerns:
>
> - **Graph size and resource**:
>
>   (1.) **Dataset property**: PartNet-Mobility(Sapien) stands as one of the largest and most widely used and compared datasets for everyday articulated object modeling. We provide statistics on the number of rigid parts from this dataset in the attached Fig.R4, where most objects have less than 8 rigid parts and the distribution has a long tail. Some extreme examples are the remote controller and the keyboard. We hope future work can address this challenge as in L313.
>
>   (2.) **Larger K and resource consumption**:  As detailed in Suppl. Tab.S2, scaling NAP’s max number of nodes from 8 to 15 and 20 still yields effective results (note that 20 nodes almost cover most objects in the dataset). Because the parameterization is compact (different from image generation) NAP can be trained extremely fast (L26 in Supp): when the max number of nodes K=8, the training batch size=64, and supervise 10 diffusion steps in a forward, we only have to consume 8465MB vRAM. The training can be done in 9 hours on one RTX3090 GPU. For inference: when set K=20, generating 10 objects in parallel on an RTX3080-Laptop GPU only takes 20 seconds with 2545MB vRAM cost.
> - **Articulated objects claim**: Yes, there is a huge literature on humans and animals, but this paper aims to focus on synthesizing, specifically, the daily articulated objects. We can change our title from “Neural 3D Articulation Prior” to “Neural 3D Articulated Object Prior” and explicitly emphasize the scope precisely in the paper. While humans and animals usually have some template parametric models, it’s very interesting to study how to generate these deformable objects. We will update our related work section with more literature about humans and animals in our revision.
> - **Why is latent diffusion worse**? As in L259, latent diffusion first generates a latent code of the object. However, since the decoder is learned, maybe a slight error made by the generated code in latent space will lead to the very wrong decoded structure in the 3D space. On the contrary, NAP directly diffuses the whole graph and its attributes in the 3D space so the errors will be controlled better from the loss. Another advantage is that all the convenience and flexibility of conditional generation come from this explicit graph space diffusion.
> - **Assumptions as limitations**: Thanks, we will append a new item to the limitation. As in the attached Fig.R5, two examples violating the assumptions are shown. The left object has a chain in the middle, which can be converted to a tree with one dummy node plus motion synchronization; the Airbus handle on the right violates the screw assumption where the joint can have two revolute DoF, which can also be resolved by adding a dummy node for the second DoF. We leave for future works to study these more complex kinematics.
> - **Why Plucker?** Plucker is not the only way to represent joints but an elegant and simple way. As studied in [56], such representation unifies the revolute and prismatic joints and avoids the redundancy caused by defining additional joint coordinate systems (e.g. moving the joint frame along the joint axis still represents the same axis).
> - **Clarification, writing, and format**: Thanks for all these careful notes and checks! We take all these seriously. Please see also the global response [G1]:
>   - *L109:* Yes $T_{gi}$ is a node property, which is computed from the joint initial state ($\theta=0,d=0$).
>   - *L112:* The $\mathbb R^3$ means the concatenation of three real numbers, representing the lengths of the bbox.
>   - *L115:* The subscript $i$ for $o_i$ means for one node, which either exists or does not, but if we concatenate all nodes together then $o$ should be in {0,1}$^K$.
>   - *L116:* Yes, sorry for this typo, $f_i$ should be $s_i$
>   - *L121:* The $\mathbb S^2$ means a unit sphere, which means that $l$ should be a unit-length vector.
>   - *L124:* L124 is to motivate defining the joint parameters in the global frame. For the same joint, its parameters written in the child and the parent frames are quite different (since the axis line is located differently). If defined locally in either the child or parent frame, when the parent and child node order flips, the edge has to change significantly, which may be harmful to the later learning process.
>   - *L125:* Yes, it can be negative
>   - *L128-1:* $\theta$ means the revolute joint angle and $d$ is for prismatic displacement.
>   - *L128-2:* Sorry for this confusion, we realized that the text definition of $T(\theta, d)$ is not precise. Let’s say i is the parent part and j is the child part. $T(\theta, d)$ is the motion of the child part from its initial rest state caused by joint states $(\theta, d)$ expressed in the parent frame. So $(\theta, d)$ can be regarded as defined in the reference of parent i. This is illustrated in the attached Fig.R8 (right) and we will append a detailed text explanation to our revision.
>   - *L129-1:* Here the subscript means: written in which coordinate frame. $(l_g, m_g)$ is the plucker coordinate of the joint written in the global object frame.
>   - *L129-2:* Yes, $T_{gi}=[R_{gi};t_{gi}]$, we’ll clarify our subscripts convention as in L110 in our revision.
>   - *L136-1:* We represent the unit vector $l$ with 3 numbers, and $p_{i,j}$ has a dimension of 6.
>   - *L136-2:* Will update this. But note that during prediction, the network still predicts a continuous scalar.
>   - *L146:* To better link to the diffusion formula (all stuff as one $x$ in later equations), we can change the $+$ to, for example, $\oplus$.
>   - *L155/Eq.4:* We will change the notation.
>   - *Occupancy or SDF?:* The SDF is used for prediction and supervision, but the network architecture is an OccNet encoder-decoder.

---

> > ### Comment · Reviewer_ncuk · 2023-08-19
> > **Response to rebuttal**
> >
> > First of all, thanks for the detailed rebuttal, not only to answer my questions, but also to address the doubts from the other reviewers. There are quite a lot reviews for this work and many of the questions are properly discussed. I have read the entire reviewing thread and I appreciate all the efforts from the authors.
> >
> > I agree with the others that this paper studies an interesting problem through introducing LDM into articulated object generation. This is why I'm totally fine if this work goes into the final proceeding. At this time point, I choose to keep my original rating since my concern regarding the writing and claims remain without seeing the improved version.

---

> > > ### Author Response · Authors · 2023-08-20
> > > **Thank you!**
> > >
> > > Thanks for your feedback and we really appreciate your engagement in the discussion!
> > >
> > > NeurIPS guidelines do not allow us to post a complete improved revision of the paper.
> > > However, we share in the following a revised draft of two paragraphs from Sec.3.1--parameterization (from L106 - L143). Hope this would help to address your concerns. If you have any questions or suggestions, please let us know, thank you!

---

> > > > ### Author Response · Authors · 2023-08-20
> > > > **Revised draft of two paragraphs**
> > > >
> > > > **Please note that the paragraphs below are drafts and are not the final version, which is under active refinement. Also, the latex math may not be correctly or fully rendered in this block.  We move the discussion about $T(\theta, d)$ in the edge section to the supplementary material to simplify the presentation since these are details and we believe this change will enhance the readers' comprehension without compromising the clarity of our method.**
> > > >
> > > > **Nodes**:  As shown in Fig.2-top, we represent each rigid part as one node in the tree. A node captures its rigid part shape by (1) a shape latent code $s_i\in \mathbb{R}^F$ (with $F$ being channels, $i$ being the part index) that can decode the SDF[100] of the part surface, and (2) the bounding box edge lengths $b_i \in \mathbb{R}^3$ that can scale the decoded SDF properly into the part's scale. We pre-train an SDF shape Auto-Encoder[10,100] for obtaining $s_i$,  further details are provided in our supplementary material. Importantly, one should also specify how to assemble these parts into an object before further modeling the articulation motion. We do so by adding to the node attributes a part initial rigid transformation $T_{i}\in SE(3)$ in the global object frame representing the part initial configuration. $T_i$ comprises an axis-angle rotation and a translation, which can be written as a $6$-dimensional vector. As articulated objects can be instantiated to different configurations so any of these configurations can be used as $T_{i}$, in practice, we consistently choose $T_{i}$ by exploiting the canonicalization in the large annotated dataset: we use the part poses in their rest states (zero joint angles and displacements) for $T_i$. Since we observe current datasets often align their rest states consistently to static un-articulated objects, for example, PartNet-Mobility[72] aligns with static PartNet[71]. To model the varying number of parts across objects, we define a maximum number of $K$ parts as well as a per-part binary indicator $o_i\in \{0,1\}$ of part $i$ existence. In summary, a node $i$ has an attribute vector $v_i=[o_i, T_i, b_i, s_i]$ with dimension $D_v=1\oplus 6 \oplus3 \oplus F$ ($ \oplus$ means concatenation), and the overall node component of an object graph is a feature of dimension $K\times D_v$.
> > > >
> > > > **Edges**: The edge in the graph represents the motion constraint of each articulation joint. A joint possesses a 3D axis (a directed line), around which the revolute joint can rotate and along which the prismatic joint can translate.
> > > > Inspired by [56], such a joint axis is represented by Pl\"ucker coordinates $(l\in \mathbb S^2,m\in \mathbb R^3)$. Here the 3D direction vector $l$ is from a unit sphere $\mathbb S^2$ and the momentum $m$ is perpendicular to $l$. This representation avoids defining additional local joint coordinate frames with ambiguity (See supplementary material for details). Such 6-dimensional Pl\"ucker coordinates $p_{(i,j)}=(l_{(i,j)},m_{(i,j)})$ for the joint from part $i$ to $j$ is defined in the global object frame when parts are in their initial rest configuration $T_i, T_j$. To fully define the joint motion constraint, we also incorporate two joint state ranges $r_{(i,j)}\in \mathbb R^{2\times2}$ for both the prismatic translation and the revolute angle components (the left limit of the range can be negative). A purely prismatic joint will have its revolute component range set to $[0,0]$ and vice versa.
> > > > Following the nodes padding, we also pad the edges to a complete graph and use an indicator $c_{(i,j)}$ denoting the edge existence. Note that the above joint axis and range have a parent-child direction from part $i$ to $j$. When the direction flips, a notable benefit of expressing $p_{(i,j)}$ in the global object frame, rather than the local part frame, is the inherent relationship $p_{(i,j)} = -p_{(j,i)}$ and $r_{(i,j)} = r_{(j,i)}$, which motivates us to model the padded graph with only $K(K-1)/2$ edges for all $i<j$ pairs. Since the nodes have no specific order, to avoid the sign flipping in $p_{(i,j)} = -p_{(j,i)}$ when permuting the nodes and to help the network learn a more stable prior, we explicitly embed the sign inside $p_{(i,j)} = -p_{(j,i)}$ into the edge existence indicator $c_{(i,j)} \in \{-1,0,+1\}$. Here $0$ indicates an edge non-exist and $+1, -1$ indicates the chiralities of existing edges, leading to $\hat p_{(i,j)} = \hat p_{(j,i)}$ (where $\hat p$ denotes the actual prediction target).
> > > > In essence, each edge is characterized by an attribute vector $e_{(i,j)} = [c_{(i,j)}, p_{(i,j)}, r_{(i,j)}]$ with a dimension of $D_e=1 \oplus 6 \oplus 4$. The overall edge parameterization of an object has shape $K(K-1)/2\times D_e$.
> > > >
> > > > We will see later in Sec.3.2 how to extract an articulated object model from the diffusion model prediction via the Minimum Spanning Tree and more details including how to compute the articulation motion from the above parameterization are in our supplementary material.

---

### Official Review · Reviewer_pAau · 2023-07-10

**Soundness:** 4 excellent
**Presentation:** 4 excellent
**Contribution:** 4 excellent
**Rating:** 7
**Confidence:** 3

**Summary:**

The paper presents the first generative model over articulated shapes. The paper presents several contributions: (1) a parameterization of articulated shapes that are easy to be used with neural networks, (2) design of denoising diffusion architectures that are structured and can effectively denoise the shapes in the introduced parameterization, and (3) extensive evaluations using novel and meaningful metrics.

**Strengths:**

The paper presents the first such generative model of articulated shape, and demonstrates excellent results. The extensive baseline comparisons support the claims. While latent diffusion comes close in performance, I believe the structured models introduced in this paper would become even more relevant with more complex shapes.

The ideas presented are original and are very well explained. I appreciated the supplemental video visualizations.

**Weaknesses:**

- The model is trained on a dataset of around 2k shapes. I wonder if there is significant overfitting. Visualizing the nearest training shape from the generated samples would be helpful.
- The forward process adds noise to discrete indicator variables (both for nodes and edges). How is the binning done during sampling, i.e., from a floating point value, how is the indicator value computed?
- Adding noise to the introduced parameterization leads to intermediate noisy states that do not correspond to any valid 3D shape. Is that true? If not, it would be great to add a visualization of the diffusion process.
- The baseline details are not all obvious. The technical exposition is at a high level. Without publicly available code, these evaluations would not be reproducible.
-  How is the random sampling over joint poses done for implementing the ID metric? Is the distribution over poses known for each shape?

**Questions:**

Listed in weaknesses.

**Limitations:**

Limitations are adequately discussed.

---

> ### Author Rebuttal · Authors · 2023-08-08
>
> Thanks for your feedback! Here are our responses to your questions and comments and we hope that they could help to address your concerns:
>
> - **Overfitting**: Please see [G2] in our global response block.
> - **Indicator variables**: We apologize for any confusion caused. As detailed in Suppl. L39, we initially determine node existence by thresholding the node indicator o at 0.5. For all identified existing nodes (which form a sub-complete graph), we utilize the edge indicator/chirality c value, specifically -|c|, to construct a Minimum Spanning Tree (MST). The binarization process occurs within the MST during the selection of tree edges. We will clarify this procedure in our revised manuscript.
>  - **Diffusion process**: Yes, when adding noise, for example, to the plucker coordinates, the intermediate steps may not correspond to valid joint parameters anymore. While we currently treat the parameterization as if it exists in continuous Euclidean space and employ standard Euclidean diffusion (which has proven effective), we concur that leveraging advanced manifold diffusion techniques (like Grassmannian for Plucker) or discrete diffusion methods for indicators could enhance the generation process. This point is highlighted in our limitations section (L316). We also visualize the intermediate steps by projecting the noisy parameterization back to the nearest valid parameterization, the bottom three small figures in Fig.2 and the small animation in our video starting from 2:51 are these visualizations. We will enhance the clarity of these visualizations in our revised version.
> - **Baselines**: We provide more details of the baselines in Suppl. L53 Sec.S1.2, and we will release the code once the paper is published. Hope these baselines would also help the new area.
> - **ID Pose sampling**: As outlined in L222, each joint possesses working ranges (predicted for generated objects and sourced from ground truth for references) for both prismatic and revolute modes. We uniformly sample within these ranges to obtain the sampled joint states, implying that the distribution is simply a uniform distribution, parameterized by the range's endpoints.  We will make this more clear in our revision.

---

> > ### Comment · Reviewer_pAau · 2023-08-15
> >
> > Thank you for the rebuttal. It answers all my questions. The nearest neighbors seem very close to the generated samples, implying that the model does not generalize / interpolate much. This could be due to the scale of the training data, as the authors mention. I believe the method and the results still provide useful insights.

---

> > > ### Author Response · Authors · 2023-08-15
> > > **Thank you!**
> > >
> > > Thanks for your feedback and we really appreciate your time!
> > >
> > > Another potentially exciting direction we are recently thinking about is to learn such articulated object priors from large-scale image or video datasets, from where we may hybrid the knowledge in accurate but expensive smaller URDF datasets (like PartNetMobility) and large-scale but unlabelled video/image dataset.

---

### Official Review · Reviewer_zyR9 · 2023-07-29

**Soundness:** 3 good
**Presentation:** 2 fair
**Contribution:** 3 good
**Rating:** 7
**Confidence:** 3

**Summary:**

This paper introduces the novel task of articulated 3D object generation. The method devises a parameterization of an articulated 3D object by representing it as a complete graph with nodes corresponding to parts and edges corresponding to joints. The method then trains a diffusion model on this parameter space. Once parameters are generated with the diffusion model, they can be converted back to an articulated 3D object. This paper also proposes a novel metric for evaluation that considers both the geometry and motion of the object. The paper compares to adapted baselines both quantitatively (using the new metric) and qualitatively. Results suggest that this approach outperforms the adapted baselines.

**Strengths:**

1) Novel task. NAP solves the task of articulated 3D object generation. The task is useful for creating data that incorporates motion and this work could inspire future research in this new area.

2) This paper devises a parameterization of 3D articulated objects that can be integrated with a diffusion model.

3) The paper takes a thorough approach to evaluation creating a novel distance metric that considers both geometry and motion. Results outperform adapted baselines both quantitatively and qualitatively.

**Weaknesses:**

1) This paper could benefit from more clarity on distinctions between contributions from this work versus existing techniques. Specifically, consider adding more detail differentiating the parameterization contributions of this paper that are distinct from prior work as it seems that the parameterization is closely based on URDF [1] for converting the articulated object into the graph representation that can then be used with diffusion. Additionally, further distinction between general graph-attention denoising networks and the contribution of this work’s architecture would be helpful.

2) This method relies heavily on existing techniques such as diffusion models and graph representations of articulated objects. Significant contribution seems to come from the novelty of the task itself and the way in which the paper combines these existing techniques to solve it.

Minor comments:
Figure 4 showing the qualitative comparisons is slightly confusing. More labels on the image itself would be helpful for clarity.

References:
[1] Morgan Quigley, Brian Gerkey, and William D Smart. Programming Robots with ROS: a practical 407 introduction to the Robot Operating System. " O’Reilly Media, Inc.", 2015.

**Questions:**

Please further clarify the contributions regarding representation parameterization and graph-attention architecture as compared to existing methods. My understanding for the parameterization is that the novelty comes from choosing what features need to be contained in nodes (indicator, pose, bbox, shape code) and edges (chirality [indicator], Plucker, joint limits).

**Limitations:**

Authors discuss the limitations of their work.

---

> ### Author Rebuttal · Authors · 2023-08-08
>
> Thanks for your feedback! Here are our responses to your questions and comments and we hope that they could help to address your concerns:
>
> - We agree that the introduction of a new task and the establishment of a benchmark metric are pivotal contributions of this paper. We also make the first attempt to provide a deep-learning solution for the articulated object synthesis problem:
>
>   - **Parameterization/Representation contribution**: The URDF format, widely adopted for modeling robots and articulated objects, served as an inspiration for our representation. However, it's crucial to understand that our representation is tailored for neural network compatibility – designed to be fed into and extracted from a learned model, ensuring it can seamlessly represent a diverse range of different structured articulated objects in a dataset. While URDF is intuitive, our representation has several differences:
>
>     - (1.) We define the joint and part relationships in a more unified and elegant way: in URDF, one must specify the joint type as revolute or prismatic and define the parent-to-joint and optionally the joint-to-child transformation between two parts, which leads to redundancy (e.g. translating joint coordinate frame along the joint axis won’t change the joint). Instead, we model the joint in the global object coordinate system with screw representations and use per part pose in the global coordinate frame to define the initial relative transformations between the parent and child, which unifies the revolute and prismatic joint and is more stable and cleaner for the network to learn.
>     - (2.) We pad the tree to a complete graph, adding indicators to the complete graph, and use MST postprocessing to extract back the tree from the network output. With all these careful designs of representation that do not come with URDF, we enable the processing of deep networks on this highly irregular collection of different articulated objects.
>
>      Compared to existing methods of modeling articulated objects, they either model a fixed/simple structure of articulation, a fixed number of parts (usually two), or implicitly plug the joint states into the latent code instead of explicitly modeling the structure. Our representation, inspired by URDF and with the above-mentioned difference to URDF, is designed for neural processing and is an explicit and holistic description of various structured objects. In short, the representation contribution is a complete description of various articulated objects across the dataset that is oriented to the neural network.
>
>   - **Deinosing network distinction**: Indeed, there are many graph-based networks for denoising and some diffusion models for generating large and general graphs. We leverage the general attention mechanism on graphs and tailor it to our unique task. In our problem, we pay more attention to the node and edge attributes and the key observation is that the information exchange between edge and nodes is important (joints and parts talk). Thus we utilize a graph attention network as in Sec.3.3 that explicitly fuses information of both node and edge. We also hybridize a pointnet-like global pooling over all node features to gain more global information. While we don't highlight this network as our primary contribution, its reasonable design is rooted in our deep understanding of the task at hand.
>
>   In summary, our work is not just a combination of existing methods: besides the significant contribution of our novel task, we also contribute a novel representation that is tailored to neural network diffusion and design a denoising network with our insight into this task. We hope our first attempt toward this brand-new direction may inspire future works to propose better solutions.
>
> - **Figure suggestion**: We appreciate the constructive feedback. To enhance clarity, we will incorporate labels into the sub-images in our revised version.

---

> > ### Comment · Reviewer_zyR9 · 2023-08-19
> >
> > Thanks for the rebuttal. It addresses my concerns. I still think that this paper solves an interesting new problem and could inspire future work on this topic. After reading the other reviews and the rebuttal, I am convinced of the contributions of this paper and will raise my score.

---

> > > ### Author Response · Authors · 2023-08-19
> > > **Thank you!**
> > >
> > > Thanks for your feedback and we really appreciate your time!

---

### Author Rebuttal · Authors · 2023-08-08

We thank the reviewers for their constructive comments and are glad that they all have a consensus on our contributions/novelty:

1. **Contribution and novelty of studying a new problem**
2. **Contribution of new evaluation metrics to this novel task**
3. **Technical contribution of the first deep-learning framework on this problem**
4. **Performance and effectiveness of our approach including novel applications**

We believe that, despite some detailed imperfections, our work has the potential to inspire future works and pave the way for new research areas.

We will answer the review comments and questions and hope this could help address some concerns. Here we will first respond to some common questions in this block and then provide individual responses to each reviewer. To streamline the discussion, we will reference this global response in the individual sections when necessary. An accompanying PDF with figures for this rebuttal is also attached.

**Global response to common questions:**

- **[G1]** **Paper writing**: We are thankful to the careful and constructive feedback from all the reviewers including typos, inconsistencies, missing details, figure labels, and table layout. Indeed, our presentation, especially in Sec.3.1, is not that easy to follow due to the compression of the original manuscript to fit the page limits. We will revise paragraphs, update all the issues the reviewers bring up, and check all the text carefully. The discussions in this response will also be incorporated into our revised paper. As suggested by WYGQ, a draft schematic visualization of the variables described in Sec. 3.1 is provided in Fig.R8 in the attached PDF.

- **[G2]** **Overfitting**(PAAU, HNRT): Regarding this problem, we would like to argue the following points:
  1. We retrieve the nearest training object (w.r.t. our ID metric) and visualize some samples in the attached Fig.R1. As it shows, both geometric and structural differences can exist in the (generated objects, nearest neighbors) pairs.
  2. We mainly demonstrate our effectiveness in modeling articulated-object distributions because this is the most important feature of generative models. Other baselines, on the contrary, cannot efficiently fit the distribution with similar or even larger network capacities.
  3. The overfitting problem is also an active research problem for diffusion models in general [1]. Even the well-known image diffusion models show overfitting properties, yet they are widely used for their effectiveness in many applications – this also applies to our method. For example, in Sec.4.4 we show that the learned articulation prior can be used for articulating static models (Fig.6) or doing part-to-articulated-object completion with multiple proposals using conditioned generation.
  4. One bottleneck here compared to the widely-existing image diffusion models, as pointed out by HNRT, is the size of the dataset for training. Potentially, collecting larger datasets would be very beneficial to the community. On the other side, new techniques that learn articulation prior from images or videos may also help to address this issue.
  5. For evaluation, we have different training and testing dataset splits, ensuring the evaluations of the learned distribution are valid.

[1] Carlini, Nicholas, et al. "Extracting training data from diffusion models." arXiv preprint arXiv:2301.13188 (2023).

---

### Decision · Program_Chairs · 2023-09-21

**Decision:**

Accept (poster)

**Comment:**

This paper initially received mixed reviews with concerns revolving around the novelty of the task definition, some limitations of the approach not being clearly explained, concern that the presented results may not generalize due to over-reliance on a specific dataset, and lastly some clarity issues with the writing in the paper.  During the rebuttal and discussion period, the authors responded to these concerns and largely alleviated them.  With the exception of one borderline-negative reviewer who is not strongly opposed, all reviewers would like to see the paper accepted.  The AC finds no basis to overturn this near-consensus and therefore recommends acceptance.

The reviewer who remained negative (ncuk) had as a primary concern issues with the clarity of exposition.  The authors responded with rewritten parts of the paper that demonstrate a commitment to appropriately revising the draft to incorporate reviewer comments and suggestions.  The AC strongly recommends a thorough editing pass when preparing the camera ready version to incorporate the clarifications and improvements from this discussion period.